# Discrete Audio Tokens: More Than a Survey!

Pooneh Mousavi*†[1,2], Gallil Maimon*[3], Adel Moumen*[4], Darius Petermann*[5], Jiatong Shi*[6], Haibin Wu*[7], Haici Yang*[5], Anastasia Kuznetsova*[5], Artem Ploujnikov[8,2], Ricard Marxer[9], Bhuvana Ramabhadran[10], Benjamin Elizalde[11], Loren Lugosch[11], Jinyu Li[7], Cem Subakan[12,2,1], Phil Woodland[4], Minje Kim[14], Hung-yi Lee[13], Shinji Watanabe[6], Yossi Adi[3], Mirco Ravanelli[1,2,8]

[1]Concordia University, [2]Mila-Quebec AI Institute, [3]The Hebrew University of Jerusalem, [4]University of Cambridge, [5]Indiana University, [6]Carnegie Mellon University, [7]Microsoft, [8]Université de Montréal, [9]Université de Toulon, [10]Google, [11]Apple [12]Laval University, [13]National Taiwan University, [14]University of Illinois at Urbana-Champaign

## Abstract

Discrete audio tokens are compact representations that aim to preserve perceptual quality, phonetic content, and speaker characteristics while enabling efficient storage and inference, as well as competitive performance across diverse downstream tasks. They provide a practical alternative to continuous features, enabling the integration of speech and audio into modern large language models (LLMs). As interest in token-based audio processing grows, various tokenization methods have emerged, and several surveys have reviewed the latest progress in the field. However, existing studies often focus on specific domains or tasks and lack a unified comparison across various benchmarks. This paper presents a systematic review and benchmark of discrete audio tokenizers, covering three domains: speech, music, and general audio. We propose a taxonomy of tokenization approaches based on encoder-decoder, quantization techniques, training paradigm, streamability, and application domains. We evaluate tokenizers on multiple benchmarks for reconstruction, downstream performance, and acoustic language modeling, and analyze trade-offs through controlled ablation studies. Our findings highlight key limitations, practical considerations, and open challenges, providing insight and guidance for future research in this rapidly evolving area. For more information, including our main results and tokenizer database, please refer to our website: `https://poonehmousavi.github.io/dates-website/`.

## 1 Introduction

Audio compression has been a well-established research topic since the foundations of digital communication (Shannon, 1948; Nyquist, 1928). Traditional audio codecs, such as linear predictive coding (LPC) (Itakura, 1968; Atal, 1970), modified discrete cosine transform (MDCT) (Wang & Vilermo, 2003), and Code Excited Linear Prediction (CELP) (Schroeder & Atal, 1985; Jage & Upadhya, 2016), were designed to reduce redundancy and remove perceptually irrelevant information. These models have been effective in compressing raw audio signals into compact bitstreams (encoding) and then restoring them to the original signal domain (decoding). Codecs like USAC (Quackenbush, 2013), Opus (Valin et al., 2012), and EVS (Dietz et al., 2015) combine these techniques to support a range of content types, bitrates, and sampling rates while ensuring low latency for real-time communication. These approaches rely heavily on domain knowledge, combining signal processing pipelines with hand-crafted components to achieve efficient but lossy compression.

Traditional codecs are efficient and optimized for perceptual quality, but their design requires substantial manual effort, including parameter tuning and subjective listening tests (Valin et al., 2012; Dietz et al., 2015). This has motivated a shift toward data-driven approaches with deep learning, known as neural codecs. Neural codecs consist of an encoder, decoder, and a quantization module, closely resembling standard

---

†Project lead, Corresponding author (pooneh.mousavi@mail.concordia.ca)
*Equal contribution, Core Team.

autoencoders. The key difference is that neural codecs produce discrete representations (audio tokens) instead of continuous ones. The discretization is performed by a differentiable quantizer, such as residual vector quantization (RVQ) (Zeghidour et al., 2021), which enables end-to-end training by allowing gradients to propagate through the quantization step. Neural codecs are often trained using a combination of losses. For example, reconstruction losses in the time and frequency domains (Kankanahalli, 2018), optionally combined with psychoacoustic calibration (Zhen et al., 2020), direct guide signal reconstruction. Adversarial losses (Zeghidour et al., 2021) and generative models (Kleijn et al., 2018; Valin & Skoglund, 2019b; Gârbacea et al., 2019) indirectly improve the perceptual quality of the reconstructed signal. Finally, auxiliary losses are often introduced to improve the learning process and often act as regularizers or encode inductive bias (Zhang et al., 2024a; Défossez et al., 2024; Har-Tuv et al., 2025).

Discrete tokens have several useful properties. As they are normally compact, audio tokens enable more efficient storage and transmission than continuous embeddings. They also simplify audio generation by converting tasks that involve modeling continuous distributions, such as regression, into discrete classification problems (Wu et al., 2024f; Mousavi et al., 2024a). More importantly, they help bridge the gap between text and audio processing, making them a natural choice for multimodal models and a core component of many recent multimodal LLMs (Peng et al., 2024; Cui et al., 2024; Ji et al., 2024a; Latif et al., 2023; Liu et al., 2023; Wu et al., 2024a; Tian et al., 2025). Driven by these advantages, discrete audio tokens have already been adopted as an alternative to continuous features in a wide range of downstream tasks: automatic speech recognition (Chang et al., 2023; Du et al., 2023), speech-to-speech translation (Popuri et al., 2022; Inaguma et al., 2023; Wu et al., 2023a; Chang et al., 2024), voice conversion (Maimon & Adi, 2023; Wang et al., 2024d), text-to-speech synthesis (Ju et al., 2024; Chen et al., 2025a; Hayashi & Watanabe, 2020), speech enhancement (Wang et al., 2024e; Yang et al., 2024f; Xue et al., 2024), and source separation (Shi et al., 2021c; Erdogan et al., 2023; Mousavi et al., 2024b; Bie et al., 2025; Yip et al., 2024). Discrete tokens are also used in music and general audio tasks, including music generation (Copet et al., 2023; Chen et al., 2024a), environmental sound synthesis (Yang et al., 2023b; Kreuk et al., 2023), and multimodal generation (Borsos et al., 2023b; Liu et al., 2023; Ziv et al., 2024; Rubenstein et al., 2023; Wang et al., 2024b).

Recent studies have introduced a variety of tokenization methods, often grouped into two main categories: *acoustic* and *semantic*[1] (Borsos et al., 2023b; Zhang et al., 2024a; Har-Tuv et al., 2025; Guo et al., 2025b). Acoustic tokens are typically learned through encoder-decoder architectures optimized for waveform reconstruction (Zeghidour et al., 2021; Défossez et al., 2023; Kumar et al., 2023; Yang et al., 2023a). Semantic tokens are derived from pretrained self-supervised learning (SSL) models (Lakhotia et al., 2021; Mousavi et al., 2024b), which are trained on raw audio without labels by solving proxy tasks (e.g., masked prediction) to learn transferable representations to downstream tasks, or encoders trained in a supervised manner (Du et al., 2024b) designed to capture phonetic or linguistic content for discriminative tasks such as speech recognition and translation. Some recent approaches aim to combine both types, introducing hybrid tokenizers (Zhang et al., 2024a; Défossez et al., 2024) that balance acoustic and phonetic properties.

We argue the common division of discrete tokens into acoustic and semantic categories has notable limitations. Acoustic tokenizers can capture semantic information (Défossez et al., 2024; Du et al., 2023; Zhang et al., 2024a; Bai et al., 2024), while semantic tokenizers have been effectively used in generative tasks (Polyak et al., 2021; Wang et al., 2024e; Nguyen et al., 2025; Lakhotia et al., 2021; Maimon et al., 2025a; Hassid et al., 2023; Mousavi et al., 2024b; Wu et al., 2025). This overlap blurs the boundary between the two categories and suggests that the acoustic-semantic distinction alone is insufficient. Moreover, as tokenization methods continue to evolve, traditional classifications fail to capture key architectural differences and practical trade-offs. To address this limitation, we introduce a refined taxonomy that captures key design choices, including encoder-decoder, quantization techniques, training paradigms, streamability, and application domains.

Another notable gap in the literature is that existing surveys and benchmark papers have primarily focused on speech applications (Cui et al., 2024; Kim & Skoglund, 2024; Ji et al., 2024a; Anees, 2024; Guo et al., 2025b;

---

[1]It is important to clarify that the term "semantic" in the speech context does not align with its conventional linguistic meaning. In the speech context, these discrete tokens are more accurately described as phonetic units (Sicherman & Adi, 2023; Choi et al., 2024) and typically do not carry semantic content (Arora et al., 2025). In this paper, to maintain consistency across different domains (speech, audio, music) and with established terminology such as "semantic distillation," we consistently use the term "semantic."

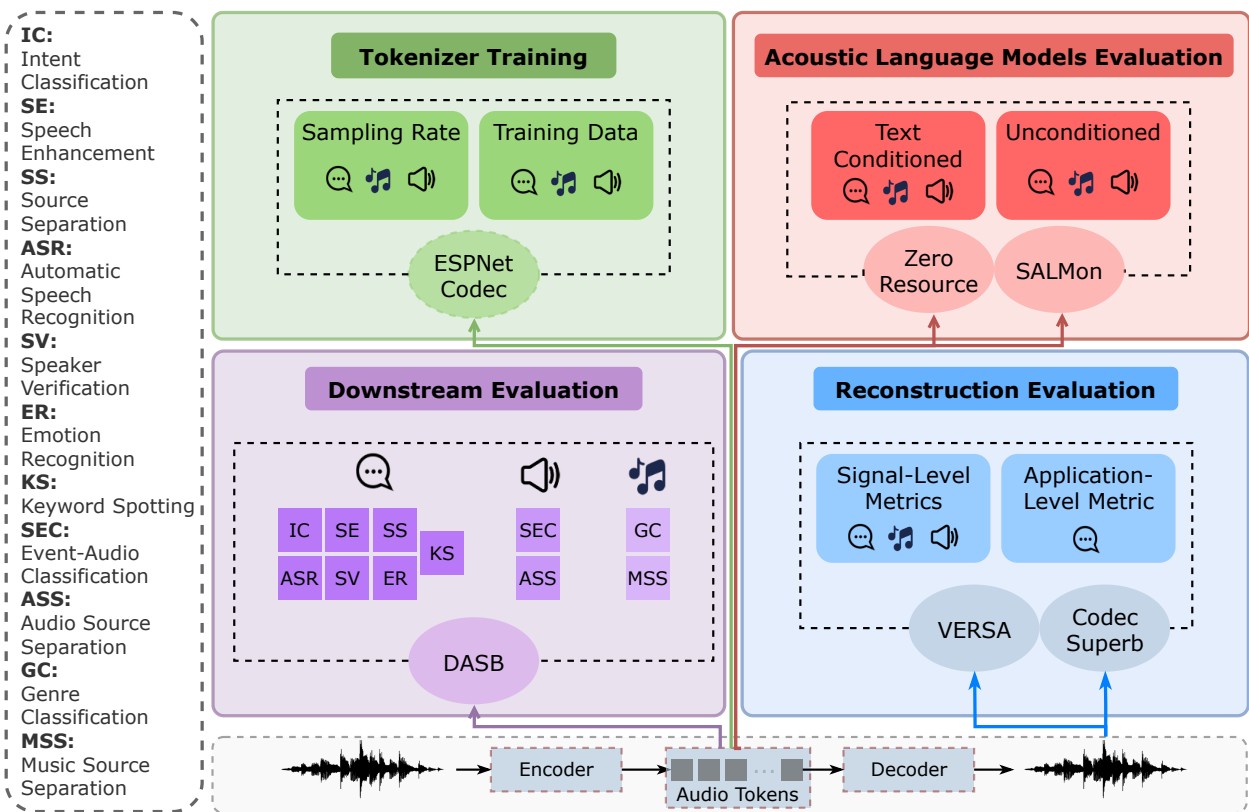

Figure 1: Overview of our empirical study, covering three domains: speech, music, and general audio, with four evaluation components: Downstream Evaluation (Section 3.2) using the DASB benchmark, Reconstructed Audio Evaluation (Section 3.1) using Codec-SUPERB and Versa, Acoustic LLM Evaluation (Section 3.3) using SALMon and the Zero-Resource benchmark, Tokenizer Training Ablation Study (Section 4) using ESPnet-Codec.

Arora et al., 2025; Vashishth et al., 2024), often overlooking tokenization methods for music and general audio. As a result, the current literature lacks a unified study that covers multiple domains and diverse evaluation criteria. Moreover, rather than providing a holistic comparison, most existing works focus on a single aspect, such as reconstruction quality in Codec-SUPERB (Wu et al., 2024c;b), downstream task performance in DASB (Mousavi et al., 2024a), controlled evaluation settings in ESPnet-Codec (Shi et al., 2024c), or audio language modeling in SALMon (Maimon et al., 2025c). These limitations persist even in the latest surveys. For example, Guo et al. (2025b) focuses on reconstruction and voice conversion, while Cui et al. (2024); Peng et al. (2024) explores integration with LLMs. To help bridge this gap, we present a comprehensive benchmark of discrete audio tokenizers. Our benchmark covers three audio domains: speech, music, and general audio. It considers multiple evaluation criteria, including signal reconstruction, downstream task performance, and acoustic language modeling. These aspects are analyzed jointly to provide a more robust and comprehensive assessment. An additional issue in current benchmarks is that tokenizers are often trained under inconsistent conditions, such as different datasets, domains, or sampling rates. These inconsistencies make direct and fair comparisons difficult. To ensure fair comparisons, we support our analysis with ablation studies that examine different quantization methods under controlled experimental settings.

Our contribution is organized into three core studies, as illustrated in Figures 1 and 3:

- **Study 1: Audio Tokenizer Taxonomy (Section 2).** We propose a comprehensive taxonomy of discrete audio tokenization methods based on key architectural and functional criteria.

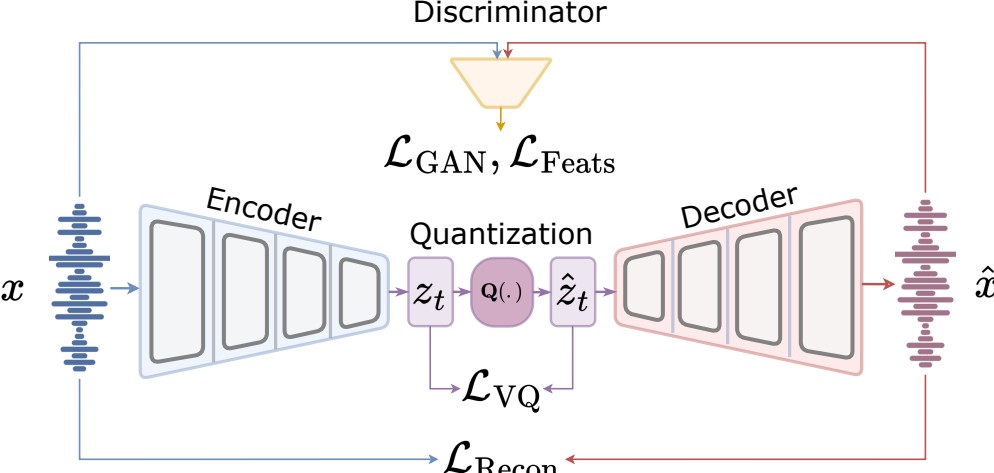

Figure 2: Overall architecture of a standard audio tokenizer. The input signal $x$ is encoded into a latent representation $z_t$, which is then discretized by a quantizer $Q(\cdot)$. The decoder reconstructs the signal $\hat{x}$ from the quantized representations $\hat{z}_t$. Training typically involves a combination of reconstruction ($\mathcal{L}_{\text{Recon}}$), adversarial ($\mathcal{L}_{\text{GAN}}$, $\mathcal{L}_{\text{Feats}}$), and vector quantization losses ($\mathcal{L}_{VQ}$).

- **Study 2: Benchmark Evaluation (Section 3).** We evaluate existing tokenizers using multiple benchmarks. Codec-SUPERB[2] and VERSA[3] (Shi et al., 2025) are used for reconstruction. DASB[4] is used for downstream tasks. SALMon[5] and the Zero-resource speech benchmark[6] (Nguyen et al., 2020) are used for acoustic language modeling. All evaluations are conducted under consistent conditions.

- **Study 3: Ablation Studies (Section 4).** We perform controlled experiments to isolate the effects of specific design choices for training audio tokenizers, including sampling rate and single-domain versus multi-domain training using ESPnet-Code[7] (Shi et al., 2024c).

This survey provides a unified and practical perspective on discrete audio tokenization and its role in speech, music, and general audio processing. We aim to clarify key design trade-offs, highlight current limitations, and offer guidance for future research in this evolving field.

## 2 Literature Review and Proposed Taxonomy

### 2.1 Overall architecture

As shown in Figure 2, audio tokenizers typically comprise three components:

- An encoder that converts the input waveform $x$ into a sequence of frame-wise embeddings $Z = \{z_t\}_{t=1}^T$ using an encoder function $f_e$, such that, $Z = f_e(x)$, where each $z_t \in \mathbb{R}^D$ is a continuous embedding at time step $t$.

- A quantization module that maps each embedding $z_t$ to a quantized vector $\hat{z}_t$ and a set of discrete indices $q_t = [q_{1,t}, \ldots, q_{M,t}]$ using a quantization function $Q$, i.e. $(\hat{z}_t, q_t) = Q(z_t)$. Here, $M$ denotes the number of codebooks used in the quantizer. The full sequence of quantized embeddings is denoted as $\hat{Z} = \{\hat{z}_t\}_{t=1}^T$.

---

[2]https://codecsuperb.github.io/

[3]https://github.com/wavlab-speech/versa/tree/main/egs/survey.

[4]https://poonehmousavi.github.io/DASB-website/

[5]https://pages.cs.huji.ac.il/adiyoss-lab/salmon/

[6]https://github.com/zerospeech/zerospeech2021_baseline

[7]https://github.com/espnet/espnet

- A decoder that reconstructs the waveform from the sequence of quantized embeddings using a decoder function $f_d$, i.e., $\hat{x} = f_d(\hat{z})$, where $\hat{x}$ is the reconstructed waveform.

To address the taxonomy issues outlined in the introduction, this section proposes a refined taxonomy based on three core dimensions: the quantization method, the encoder-decoder architecture, and the training paradigm (e.g., joint or end-to-end training, and the use of auxiliary components). We also provide more fine-grained categories, including streamability and the target domain of each tokenizer. The proposed taxonomy is illustrated in Figure 3 and the classification of existing audio tokenizers according to this taxonomy is shown in Table 1. The following subsection summarizes the most popular methods according to the proposed categorization.

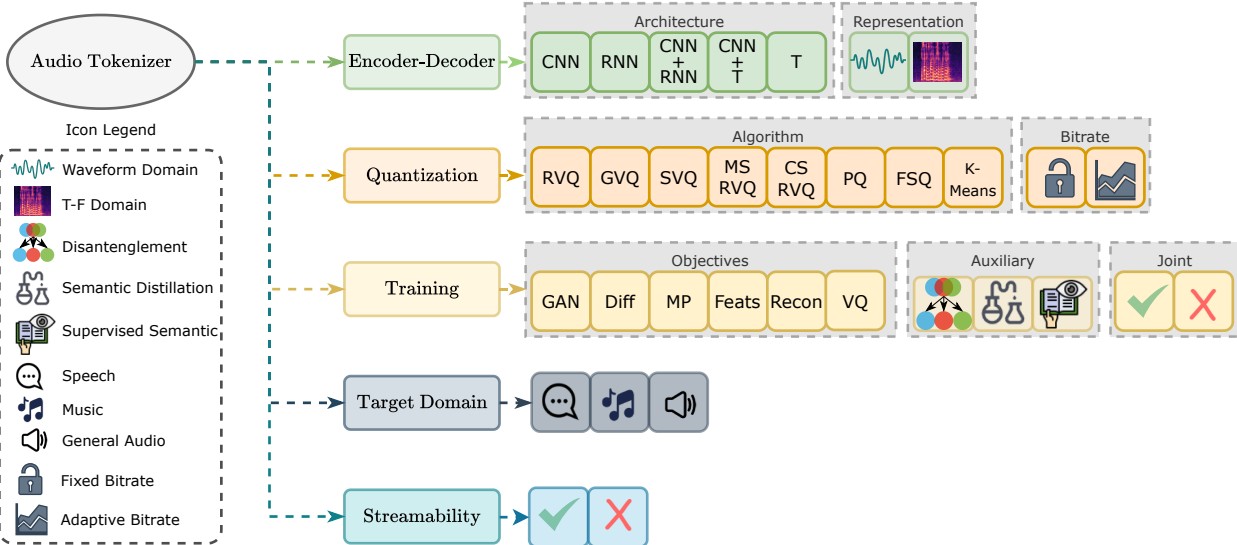

Figure 3: Taxonomy of audio tokenizers based on: encoder-decoder architecture (Section 2.3), quantization method (Section 2.2), training paradigm (Section 2.4), and target domain and streamability (Section 2.5). CNN denotes Convolutional networks, T represents Transformer models, and RNN refers to any recurrent neural network including LSTM and GRU. RVQ stands for Residual Vector Quantization, GVQ for Group Vector Quantization, SVQ for Single Vector Quantization, MSRVQ stands for Multi-Scale Residual Vector Quantization, CSRVQ stands for Cross-Scale Residual Vector Quantization, PQ stands for Product Quantization, FSQ for Finite Scalar Quantization. K-Means signifies that the tokenizer is trained independently of the encoder and the decoder pipeline. Objectives include adversarial learning (GAN), diffusion-based generation (Diff), and masked prediction (MP) as a generative training strategy, feature matching loss (Feats), and reconstruction loss (Recon). The interactive version of this figure can be accessed through https://dates-tokens.github.io/taxonomy_interactive.html

## 2.2 Quantization Method

Quantization is a key component of the tokenization pipeline, transforming continuous frame-wise features (vectors) $z_t \in \mathbb{R}^D$ into discrete tokens $q_t = [q_{1,t}, \ldots, q_{M,t}]$ and corresponding quantized vectors $\hat{z}_t \in \mathbb{R}^D$. More formally, quantization maps data into a smaller representation space with lower cardinality. It is defined as a two-step procedure involving encoding and decoding, which are distinct from the encoder and decoder modules described in Figure 2. The encoder of the quantizer, denoted $E_q : \mathbb{R}^D \to \{1, \ldots, K\}^M$, maps a continuous embedding $z_t$ to a tuple of discrete indices $q_t$, where each $q_{m,t} \in \mathcal{I}_m = \{1, 2, \ldots, K\}$ corresponds to a quantization layer (codebook) $m$. These indices refer to entries in a set of $M$ codebooks $\mathcal{C} = \{\mathcal{C}^1, \ldots, \mathcal{C}^M\}$, where each codebook $\mathcal{C}^m$ contains $K$ learnable $D$ dimensional continuous vectors. The decoder of the quantizer, $D_q : \{1, \ldots, K\}^M \to \mathbb{R}^D$, reconstructs the quantized embedding $\hat{z}_t$ by retrieving the selected codewords from the codebooks and combining them, typically via averaging or summation.

Table 1: Comprehensive overview of audio tokenizers, organized alphabetically by tokenizer name. The table covers core design choices across five major dimensions: application domains (Speech, Music, Audio), encoder-decoder architecture (including encoder/decoder type and feature representation), quantization (technique and bitrate strategy), training paradigms (objective, auxiliary loss, and joint optimization), and streaming capability.

| Tokenizer | Domain | | | Frame | Encoder-Decoder | | | Quantization | | Training | | | Stream |
| | S | M | A | Rate | Architecture | | Rep. | Tech. | Bit. | Objective(s) | Aux. | Joint | |
| | | | | | Encoder | Decoder | | | | | | | |
| APCodec (Ai et al., 2024) | ✓ | | | 150 | CNN | CNN | T-F | RVQ | F | GAN, Feat, Rec, VQ | - | ✓ | ✓ |
| AudioDec (Wu et al., 2023b) | ✓ | | | 160 | CNN | CNN | T | RVQ | F | GAN, Feat, Rec, VQ | - | ✓ | ✓ |
| Best-RQ (Chiu et al., 2022) | ✓ | | | 25 | CNN+T | - | T-F | PQ | F | MP | - | | ✓ |
| BigCodec (Xin et al., 2024) | ✓ | | | 80 | CNN+RNN | CNN+RNN | T | SVQ | F | GAN, Feat, Rec, VQ | - | ✓ | |
| DAC (Kumar et al., 2023) | ✓ | ✓ | ✓ | 75 | CNN | CNN | T | RVQ | F | GAN, Feat, Rec, VQ | - | ✓ | |
| Discrete SSL (Mousavi et al., 2024b) | ✓ | | | 50 | CNN+T | T | T | K-means | F | GAN, Feat, Rec, MP | - | | |
| Disen-TF-Codec (Jiang et al., 2023) | ✓ | | | 19 | CNN+RNN | CNN | T-F | GRVQ | A | GAN, Feat, Rec, (Pred) | Dis | ✓ | ✓ |
| dMel (Bai et al., 2024) | ✓ | | | 40 | - | CNN | T-F | SVQ | F | GAN, Feat, Rec, VQ | - | ✓ | |
| EnCodec (Défossez et al., 2023) | ✓ | ✓ | ✓ | 75, 150 | CNN+RNN | CNN | T | RVQ | F | GAN, Feat, Rec, VQ | - | ✓ | ✓ |
| ESC (Gu & Diao, 2024) | ✓ | | | 150 | T | T | T-F | CSRVQ | F | Rec, VQ | - | ✓ | |
| FACodec (Ju et al., 2024) | ✓ | | | 80 | CNN+RNN | CNN+RNN | T | GRVQ | F | GAN, Feat, Rec, VQ | Dis | ✓ | |
| FunCodec (Du et al., 2023) | ✓ | | | 1.25, 25, 50 | CNN+RNN | CNN+RNN | T-F | RVQ | F | GAN, Feat, Rec, VQ | SD | ✓ | |
| HARP-Net (Petermann et al., 2021) | | ✓ | | 44100 | CNN | CNN | T | FSQ | A | Rec | - | ✓ | |
| HiFi-Codec (Yang et al., 2023a) | ✓ | | | 50, 75, 100 | CNN+RNN | CNN+RNN | T | GRVQ | F | GAN, Feat, Rec, VQ | - | ✓ | |
| HILCodec (Ahn et al., 2024) | ✓ | ✓ | ✓ | 75 | CNN | CNN | T | RVQ | F | GAN, Feat, Rec, VQ | - | ✓ | ✓ |
| LaDiffCodec (Yang et al., 2024e) | ✓ | | | 50 | CNN | CNN | T | RVQ | F | Diff | - | ✓ | |
| Language Codec (Ji et al., 2024b) | ✓ | | | 75 | CNN+RNN | CNN | T,T-F | RVQ | F | GAN, Feat, Rec, VQ | - | ✓ | |
| LFSC (Casanova et al., 2025) | ✓ | | | 21.5 | CNN | CNN | T | FSQ | F | GAN, Feat, Rec | SD | ✓ | |
| LLMCodec (Yang et al., 2024b) | ✓ | | ✓ | 57 | CNN+T | CNN+T | T | MS-RVQ | F | GAN, Rec | SD | ✓ | |
| LSCodec (Guo et al., 2025a) | ✓ | | | 25, 50 | CNN | CNN | T | SVQ | F | GAN, Feat, Rec | Dis, SD | | |
| MDCTCodec (Jiang et al., 2024) | ✓ | | | 150 | CNN | CNN | T-F | RVQ | F | GAN, Feat, Rec, VQ | | ✓ | |
| Mimi (Défossez et al., 2024) | ✓ | | | 12.5 | CNN+T | CNN+T | T | RVQ | F | GAN, Feat, Rec, VQ | SD | ✓ | ✓ |
| MMM (Shi et al., 2024b) | ✓ | | | 50 | CNN+T | CNN | T | K-means | F | GAN, Feat, Rec, MP | - | | |
| NAST (Messica & Adi, 2024) | ✓ | | | 50 | CNN+T | CNN+T | T | FSQ | F | Rec, VQ, MP | SD | ✓ | |
| NDVQ (Niu et al., 2024) | ✓ | | | 75 | CNN+RNN | CNN+RNN | T | RVQ | F | GAN, Feat, Rec, VQ | - | ✓ | |
| PAST (Har-Tuv et al., 2025) | ✓ | | | 75 | CNN+T | CNN | T | RVQ | F | GAN, Feat, Rec, VQ | SST | ✓ | ✓ |
| PQ-VAE (Guo et al., 2024b) | ✓ | | | 75 | CNN | CNN | T-F | PQ | F | Rec, VQ | - | ✓ | |
| Prompt Codec (Pan et al., 2024) | ✓ | | | 75 | CNN+RNN | CNN+RNN | T-F | GRVQ | F | GAN, Feat, Rec, VQ | - | ✓ | |
| RepCodec (Huang et al., 2024) | ✓ | | | 50 | CNN | CNN | T | SVQ | F | Rec, VQ , MP | - | ✓ | |
| S-TFNet (Jiang et al., 2022a) | ✓ | | | - | CNN+RNN | CNN | T-F | CSRVQ | F | GAN, Rec, VQ | - | ✓ | |
| S3 (Du et al., 2024a) | ✓ | | | - | T | T | T | SVQ | F | Dif | SST | | |
| SD-Codec (Bie et al., 2025) | ✓ | ✓ | ✓ | 50 | CNN | CNN | T | RVQ | F | GAN, Rec,Feat, VQ | Dis | ✓ | |
| SemantiCodec (Liu et al., 2024a) | ✓ | ✓ | ✓ | 50 | CNN+T | T | T | RVQ | F | Dif, VQ | SD | ✓ | |
| Single Codec (Li et al., 2024) | ✓ | | | 23 | CNN+RNN | CNN+RNN | T-F | SVQ | F | GAN, Rec, VQ | - | ✓ | |
| SingOMD (Tang et al., 2024b) | | ✓ | | 50 | CNN+T | CNN | T | K-Means | F | GAN, Feat, Rec, MP | - | | |
| SNAC (Siuzdak et al., 2024) | ✓ | ✓ | ✓ | Variable | CNN+RNN | CNN | T | MS-RVQ | F | GAN, Feat, Rec, VQ | - | ✓ | |
| SOCODEC (Guo et al., 2024a) | ✓ | | | 20 | CNN | CNN | T-F | PQ | F | GAN, Rec, VQ | Dis | ✓ | |
| SoundStream (Zeghidour et al., 2021) | ✓ | ✓ | | 75 | CNN | CNN | T | RVQ | F | GAN, Rec,Feat | - | ✓ | ✓ |
| Spectral Codecs (Langman et al., 2024) | ✓ | | | 86.1 | CNN | CNN | T-F | FSQ | F | GAN, Feat, Rec | - | ✓ | |
| SpeechTokenizer (Zhang et al., 2024a) | ✓ | | | 50 | CNN+RNN | CNN | T | RVQ | F | GAN, Rec,Feat, VQ | SD | ✓ | |
| SQ-Codec (Yang et al., 2024d) | ✓ | | | 50 | CNN | CNN | T | FSQ | F | GAN, Rec | - | ✓ | |
| TAAE (Parker et al., 2025) | ✓ | | | 25 | CNN+T | CNN+T | T | FSQ | F | GAN, Feat, Rec | SD | ✓ | ✓ |
| TFNet (Jiang et al., 2022b) | ✓ | | | 120 | CNN+RNN | CNN | T-F | GRVQ | F | Rec, VQ | - | ✓ | |
| Ti-Codec (Ren et al., 2024b) | ✓ | | | 75 | CNN+RNN | CNN+RNN | T | RVQ | F | GAN, Feat, Rec, V | Dis | ✓ | |
| TS3-Codec (Wu et al., 2024d) | ✓ | | | 40, 50 | T | T | T | SVQ | F | GAN, Feat, Rec, V | - | ✓ | ✓ |
| USM (Zhang et al., 2023) | ✓ | | | 25 | CNN+T | - | T-F | PQ | F | MP | - | | |
| Vocos (Siuzdak, 2024) | ✓ | | | 50 | CNN | CNN | T-F | RVQ | F | GAN, Feat, Rec | - | ✓ | |
| WavTokenizer (Ji et al., 2024c) | ✓ | ✓ | ✓ | 40, 75 | CNN+T | CNN+T | T | SVQ | F | GAN, Feat, Rec, V | - | ✓ | |
| Wav2Vec-BERT (Chung et al., 2021) | ✓ | | | 25 | CNN+T | - | T | PQ | F | MP | - | ✓ | |
| WMCodec (Zhou et al., 2024) | ✓ | | | 75 | CNN+T | CNN | T | RVQ | F | GAN, Feat, Rec | - | ✓ | |
| X-Codec (Ye et al., 2025) | ✓ | ✓ | ✓ | 50 | CNN | CNN | T | RVQ | F | GAN, Rec, V | SD | ✓ | |
| XEUS (Chen et al., 2024b) | ✓ | | | 50 | CNN+T | - | T | K-means | F | MP | - | | |

Depending on the design choices, quantization methods vary along two important axes: (1) the specific quantization algorithm used to convert continuous features into discrete tokens, such as k-means, product quantization, or residual vector quantization (RVQ), (2) whether the bitrate is fixed or adaptive. These aspects are described in the following subsections.

### 2.2.1 Quantization Algorithm

**K-means.** K-means clustering is frequently used for post-training quantization. While many recent codec-based acoustic tokenizers tend to adopt the joint-training quantization techniques, k-means is still prevalent in extracting tokens from SSL models (Mousavi et al., 2024b; Chang et al., 2023; Polyak et al., 2021; Wang et al., 2024e). Typically, a layer or multiple layers from a pretrained SSL model are selected, and representations are clustered using offline trained k-means to create discrete tokens. Such tokenizers natively lack a built-in decoder, as they are primarily used for discriminative tasks like ASR. Nevertheless, recent studies have investigated training separate decoders to reconstruct speech from discrete representations, such as employing a modified HiFi-GAN (Yang et al., 2023a). Additionally, Mousavi et al. (2024b) introduced a multi-layer training strategy with dropout mechanisms, enabling the decoder to flexibly handle varying bitrates during inference. The assignment of each embedding $z_t$ to its nearest centroid $c_k$ is performed using the standard K-means rule:

$$q_t = \arg \min_{k \in \{1, \ldots, K\}} \|z_t - c_k\|^2 \tag{1}$$

Here, $z_t \in \mathbb{R}^D$ denotes the continuous embedding at time step $t$ from a frozen SSL model, $c_k$ is the $k$-th cluster centroid, and $q_t \in \{1, \ldots, K\}$ is the resulting discrete token index.

**Residual Vector Quantization (RVQ).** RVQ maps each frame-wise feature to the closest entry in a codebook and then refines this process by computing the residual after quantization. The remaining residual is compressed by sequentially applying a series of quantizers, each refining the residuals left by the previous one. The first neural network-based RVQ method was first introduced in SoundStream (Zeghidour et al., 2021) and has since been widely adopted in other models (Kumar et al., 2023; Défossez et al., 2023; 2024; Zhang et al., 2024a). Many approaches (Kumar et al., 2023; Défossez et al., 2023) also incorporate bitrate scalability by performing variable bandwidth training, where the number of codebooks is randomly selected during training to support different bandwidths during inference. A variant of RVQ, called Residual Normal Distribution Vector Quantization (RNDVQ), is used in (Niu et al., 2024). Unlike standard RVQ, which selects the nearest neighbor deterministically, RNDVQ formulates quantization as a probabilistic selection problem. This addresses issues such as low codebook utilization and sensitivity to noise, making the model more robust to minor variations in input data. The procedure is defined recursively as:

---

**Algorithm 1** Residual Vector Quantization (RVQ)

---

1: **Input:** Embedding $z_t$, Codebooks $\{\mathcal{C}^{(m)}\}_{m=1}^{M}$
2: Initialize residual: $r_t^{(1)} \leftarrow z_t$
3: **for** $m = 1$ to $M$ **do**
4: $\quad q_t^{(m)} \leftarrow \arg \min_k \left\| r_t^{(m)} - c_k^{(m)} \right\|^2$
5: $\quad \hat{z}_t^{(m)} \leftarrow c_{q_t^{(m)}}^{(m)}$
6: $\quad r_t^{(m+1)} \leftarrow r_t^{(m)} - \hat{z}_t^{(m)}$
7: **end for**
8: **Output:** $\hat{z}_t \leftarrow \sum_{m=1}^{M} \hat{z}_t^{(m)}$

---

Here, $z_t$ is the input embedding at time $t$, $q_t^{(m)}$ is the discrete index selected from the $m$-th codebook, and $\hat{z}_t$ is the final quantized vector produced by summing the quantized outputs from each residual stage.

**Single Vector Quantization (SVQ).** SVQ uses a single codebook for quantization, where each frame-wise embedding is mapped to a single code, unlike RVQ, which uses multiple codes. SVQ can be viewed as a simplified form of RVQ with a single codebook (i.e., $M{=}1$), without iterative residual refinement, similar to VQ-VAE (Gârbacea et al., 2019). It has gained popularity due to the architectural complexity introduced by multiple codebooks in acoustic language models, such as managing multiple codebook streams. SVQ, by contrast, is simpler and particularly useful for training acoustic language models. To compensate for the potential loss of information caused by using a single codebook, some SVQ-based codec models adopt larger codebook sizes. Examples of SVQ-based codecs include BigCodec (Xin et al., 2024), TS3-Codec (Wu et al., 2024d), WavTokenizer (Ji et al., 2024c).

**Group Vector Quantization (GVQ).** One limitation of RVQ is that most of the information tends to be captured in the first-layer codebook, with later codebooks contributing minimally. To address this, GVQ (Yang et al., 2023a) increases capacity at the first quantization stage by dividing the latent feature vector $z_t \in \mathbb{R}^D$ into $G$ non-overlapping groups:

$$z_t = \left[ z_t^{(1)} \parallel z_t^{(2)} \parallel \ldots \parallel z_t^{(G)} \right], \tag{2}$$

where each $z_t^{(g)} \in \mathbb{R}^{D/G}$ represents a segment of the input feature, and $\parallel$ denotes concatenation. Each group is quantized independently using a separate RVQ module, producing a group-wise quantized embedding $\hat{z}_t^{(g)}$. The final quantized vector is formed by concatenating the quantized outputs from all groups:

$$\hat{z}_t = \left[ \hat{z}_t^{(1)} \parallel \hat{z}_t^{(2)} \parallel \ldots \parallel \hat{z}_t^{(G)} \right]. \tag{3}$$

This grouped structure improves performance while reducing the number of required codebooks.

**Finite Scalar Quantization (FSQ).** Unlike traditional vector quantization, FSQ maps each dimension of a feature vector to a fixed set of scalar values (Mentzer et al., 2024). Specifically, each feature value $z_t$ is first squashed into the range $[-1, 1]$ using a non-linear function such as tanh, and then quantized into a scalar latent space by computing round$(z_t \cdot S)/S$, where $S$ is a hyperparameter controlling the quantization resolution. This procedure results in $2S + 1$ distinct scalar values per dimension, ensuring uniform coverage of the latent space. FSQ has been adopted in various recent models. SQ-Codec (Yang et al., 2024d) achieves this by creating a scalar latent space, while Spectral Codecs (Langman et al., 2024) use FSQ to encode mel-spectrogram features into a flat codebook. FSQ is often used with diffusion models for high-quality audio generation. HARP-Net (Petermann et al., 2021) similarly applies FSQ but directly maps bottleneck features i.e., single scalar values rather than vectors, to a set of learned scalar bins. Unlike other approaches, HARP-Net maintains the original input frame rate (44.1 kHz) by avoiding temporal decimation, instead expanding the feature dimension in intermediate layers before collapsing to scalar quantization. FocalCodec (Della Libera et al., 2025) instead uses a variant of FSQ called Binary Spherical Quantization (BSQ), which relies on two scalar values.

**Multi-Scale RVQ (MSRVQ).** MSRVQ (Siuzdak et al., 2024) extends standard RVQ by applying quantizers at different temporal resolutions. This hierarchical structure enables the model to efficiently capture both coarse and fine-grained details. The initial VQ layers operate at higher frame rates to encode fine details, while later layers work at lower temporal resolutions to refine the residuals using fewer tokens. At each stage $i$, the residual $r_t^{(i)}$ is downsampled by a factor $W_i$, quantized, and then upsampled back to length $T$:

$$\hat{z}_t^{(i)} = \text{Upsample} \left( Q^{(i)} \left( \text{Downsample}(r_t^{(i)}, W_i) \right) \right) \tag{4}$$

This strategy reduces the number of tokens while preserving essential information in the representation.

**Cross-Scale RVQ (CSRVQ).** CSRVQ (Gu & Diao, 2024; Jiang et al., 2022a) extends RVQ by integrating multi-scale features that progressively encode coarse-to-fine information. Unlike conventional RVQ, which applies residual quantization only at a single and lowest-resolution layer, CSRVQ encodes residuals between

encoder and decoder features at multiple hierarchical levels. During decoding, each level is conditioned on quantized residuals from coarser scales, allowing the model to refine reconstructions in a coarse-to-fine manner. This structure enables the preservation of low-level detail often lost in high-level-only representations. In practice, CSRVQ can include quantization modules across different decoder layers, with each layer incorporating its own quantizer. ESC (Gu & Diao, 2024) adopts this design via hierarchical transformers and stepwise decoding, progressively improving reconstruction fidelity without requiring extra fusion networks between encoder and decoder.

**Product Quantization (PQ).** Product quantization is commonly used in self-supervised learning (SSL) models to discretize continuous speech representations. PQ can be viewed as a group of independent vector quantization modules (Chung et al., 2021; Guo et al., 2024a), partitioning embeddings into smaller sub-vectors and quantizing each separately. The quantized sub-vectors are then concatenated to form the final output. Other variations include Random-Projection Quantization, as seen in models like Best-RQ and USM (Chiu et al., 2022; Zhang et al., 2023). This method maps speech signals into discrete labels using a randomly initialized projection matrix. Unlike other quantization methods, most PQ-based approaches do not have a built-in decoder, as they are primarily designed for SSL models rather than direct waveform reconstruction.

### 2.2.2 Fixed vs. Adaptive Bitrate

Depending on the system design, the bitrate of quantized representations can be either *fixed* or *adaptive*. In fixed-allocation schemes, such as those based on codebooks, the bitrate is determined by the number of bits required to represent each code index, irrespective of the actual token distribution. In contrast, *adaptive bitrate* refers to entropy-based coding schemes that assign variable-length codes based on the statistical frequency of tokens (Agustsson et al., 2017; Kankanahalli, 2018). More frequent tokens are encoded using fewer bits, while rarer tokens require more, leading to improved compression efficiency. Standard methods such as Huffman coding or arithmetic coding are commonly employed to exploit this redundancy. Importantly, any quantization method, regardless of its original design, can benefit from post-hoc entropy coding to further reduce the effective bitrate. It is also important to distinguish between *adaptive bitrate* and *scalable bitrate*. Adaptive bitrate dynamically adjusts the number of bits per token according to the token distribution (e.g., via entropy coding (Jiang et al., 2023; Petermann et al., 2021)). In contrast, scalable bitrate refers to systems capable of operating at multiple fixed bitrates, typically achieved by varying the number of active codebooks. This bitrate level is selected manually or defined as a hyperparameter, but it remains fixed per run and does not adapt token-wise at runtime. For instance, Encodec (Défossez et al., 2023) enables scalable bitrate by employing a codebook dropout strategy during training.

### 2.3 Encoder-Decoder

This section describes the main encoder and decoder architectures, along with the encoder input and decoder output representations used across different designs.

### 2.3.1 Architecture

**Convolutional (CNN).** CNN extracts and downsamples audio waveforms into lower frame-rate features using CNN layers. CNN is the most widely applied architecture among early neural codecs (Kumar et al., 2023; Zeghidour et al., 2021). CNN models are generally more compact, thus can be readily integrated with different system sizes and are especially useful in resource-constrained environments. However, CNN tokenizers cannot capture long-range dependencies.

**Convolutional + RNN (CNN+RNN).** Some tokenizers (Défossez et al., 2023; Xin et al., 2024) combine CNN-based feature extraction with LSTMs or GRUs for sequential modeling. RNN provides a mechanism for longer-range dependency than CNN, although it can still suffer from memory loss when the sequences reach a certain length. RNNs can easily add algorithm and system complexity to both training and inference. Therefore, the number of RNN layers used in tokenizers is usually small, and they are combined with CNN modules.

**Transformer (T).** This category refers to the fully transformer-based models without convolutional components (Wu et al., 2024d). This type of tokenizer is less common than others. They may achieve impressive compression and reconstruction performance, but they generally demand a large amount of training data and much heavier computational resources, which limit their practicality in tokenization.

**Convolutional + Transformer (CNN+T).** This group of tokenizers (Mousavi et al., 2024b; Chiu et al., 2022; Yang et al., 2024b) uses CNN-based feature extraction followed by attention mechanisms to capture long-range dependencies. Transformers have recently started to appear in audio tokenizers, as a replacement for RNN, due to their effectiveness in capturing long-range dependencies.

### 2.3.2 Input and Output Representations

Encoders can process audio inputs in either the time domain or the frequency domain. In the time domain approach, raw waveforms are directly passed to the encoder. In the frequency-domain approach, precomputed mel-spectrograms or other spectral features are used as inputs. The output representation can follow two approaches: (1) time domain waveforms, where the decoder directly upsamples the discrete representation into waveforms (Défossez et al., 2023); or (2) time-frequency domain features, where the decoder outputs time-frequency domain features (Siuzdak, 2024), and the Inverse Short-Time Fourier Transform (ISTFT) is applied for upsampling. In this case, the decoded features typically have a frame rate similar to that of the codec tokens. Siuzdak (2024) argues that assigning the upsampling to the ISTFT reduces the burden on the decoder and leads to better performance.

## 2.4 Training Paradigm

In this section, we discuss three key dimensions of audio tokenizer training: the training strategy, the main training objectives used to optimize the model, and the auxiliary components that further enhance the learned representations.

### 2.4.1 Training Strategies

Training strategies for audio tokenizers can be categorized into two broad approaches: *separate (post-training)* and *joint (end-to-end training)*. These differ in how the encoder, quantizer, and decoder modules are optimized in relation to each other. In both cases, the encoder may be randomly initialized or initialized using a pretrained model, such an SSL model (e.g., wav2vec 2.0 (Baevski et al., 2020), HuBERT (Hsu et al., 2021), WavLM (Chen et al., 2022)) or an ASR model (Radford et al., 2023)).

**Separate (Post-Training).** In this approach, the encoder and decoder are optimized independently from the quantization module. This is common in semantic tokenizers and earlier neural codecs. The encoder is often initialized from a pretrained SSL model such as wav2vec 2.0 (Baevski et al., 2020), HuBERT (Hsu et al., 2021), or WavLM (Chen et al., 2022), and typically kept frozen during quantizer training (Lakhotia et al., 2021; Mousavi et al., 2024b; Shi et al., 2024b). The quantizer is trained offline using methods such as k-means clustering on latent representations. Discrete SSL (Lakhotia et al., 2021; Mousavi et al., 2024b; Shi et al., 2024b), for example, uses k-means clustering to quantize the latent semantic features after pretraining. LPCNet-based codecs[8] (Valin & Skoglund, 2019a;b; Yang et al., 2023c) use a combination of scalar quantization and a multi-stage vector residual quantization on the pre-extracted features (pitch and cepstra) with k-means, involving different levels of feature predictions. LAST (Turetzky & Adi, 2024) uses VQ to quantize adapted SSL features, with the objective of improving SpeechLM, and then separately trains a HiFi-GAN based vocoder. $\mu$-law is also a popular quantization technique used in earlier autoregressive vocoders (van den Oord et al., 2016), and some neural audio codecs (Kleijn et al., 2018). Decoders are then trained independently to reconstruct waveforms or features from discrete tokens, often using HiFi-GAN (Polyak et al., 2021; Kong et al., 2020) or diffusion models (Du et al., 2024b; Zeng et al., 2024).

---

[8] https://github.com/xiph/LPCNet

**Joint (End-to-End Training).** In joint training, the encoder, quantizer, and decoder are optimized simultaneously within a unified end-to-end framework. This approach is commonly adopted by acoustic tokenizers (Zeghidour et al., 2021; Défossez et al., 2023). The full model is optimized using a combination of reconstruction losses (e.g., MSE) and often adversarial losses (Goodfellow et al., 2020) to promote both signal fidelity and perceptual quality. To address the non-differentiability of quantization, several gradient approximation techniques are used: (1) straight-through estimators (van den Oord et al., 2017), which copy gradients across the quantizer; (2) soft-to-hard quantization with annealing (Agustsson et al., 2017; Kankanahalli, 2018); and (3) Gumbel-softmax relaxation (Jang et al., 2017; Maddison et al., 2017). Joint training also allows for incorporating auxiliary objectives (see Section 2.4.3) to improve downstream task utility, robustness, or bitrate flexibility (Niu et al., 2024; Kumar et al., 2023).

### 2.4.2 Main Training Objectives

Audio tokenizers are optimized using different main objectives, depending on the targeted application, as depicted in Figure 2.

**Reconstruction (Recon).** The most common objective for training audio tokenizers is to reconstruct the original audio input from discrete tokens. This is achieved using a regression loss, such as the mean squared error (MSE) or mean absolute error (MAE) between the input $x$ and the reconstructed output $\hat{x}$ (Défossez et al., 2023; Zeghidour et al., 2021):

$$\mathcal{L}_{\text{Recon}} = \sum_{t=1}^{T} \|x_t - \hat{x}_t\|^2. \tag{5}$$

**Vector Quantization (VQ).** In the straight-through estimator (van den Oord et al., 2017) used for vector quantization, gradients bypass the codebook, requiring additional losses to align the embeddings with the encoder outputs. One example is the soft-to-hard scheme (Agustsson et al., 2017), where a quantization loss is applied during training to encourage the softmax-based quantization approximation to closely match the original continuous representation $z$:

$$\mathcal{L}_{VQ} = \|z - \hat{z}\|, \ \hat{z} = \sum_{m=1}^{M} \alpha_m * c_m, \tag{6}$$

where $M$ denotes the total number of codebooks, $c_m$ is the continuous representation that corresponds to the $m$'th codebook (as defined in Section 2.2), and $\alpha_m$ are their corresponding softmax weights.

Another example is the commitment loss, which encourages the encoder outputs to align with the selected codebook embeddings. The loss is computed between each residual $z_t^{(m)}$ and its quantized counterpart $\hat{z}_t^{(m)}$ from the $m$-th codebook, with gradients blocked from flowing through the quantized values:

$$\mathcal{L}_{\text{VQ}} = \sum_{t=1}^{T} \sum_{m=1}^{M} \left\| z_t^{(m)} - \text{sg}\left[\hat{z}_t^{(m)}\right] \right\|^2, \tag{7}$$

where sg[·] denotes the stop-gradient operator. This loss penalizes discrepancies between the encoder outputs and their corresponding quantized embeddings while ensuring that gradients do not update the codebook entries directly. Modern approaches often replace this loss with Exponential Moving Average (EMA) updates for the codebook, which improve training stability and mitigate codebook collapse. This collapse occurs when the model selects only a few code vectors, leaving the rest inactive and unupdated (Dhariwal et al., 2020; Kumar et al., 2023). As a result, the effective codebook size decreases, lowering the target bitrate and degrading reconstruction quality. To address this issue, researchers have proposed several strategies to improve codebook utilization. Some models (Dhariwal et al., 2020; Zeghidour et al., 2021) apply codebook expiration, periodically reinitializing inactive code vectors. Others (Kumar et al., 2023; Yang et al., 2024d; Défossez et al., 2023) use factorized quantization and L2 normalization to encourage more balanced usage across the codebook. Techniques like ESC (Gu & Diao, 2024) and NDVQ (Niu et al., 2024)introduce

Euclidean normalization or represent codebooks as distributions, using margin-based or probabilistic losses to prevent collapse and promote diverse activation. Additional solutions introduce auxiliary constraints, such as entropy penalties or code balancing losses. For example, Enhanced Residual Vector Quantization (ERVQ) Zheng et al. (2025) uses intra-codebook optimization via online clustering and a code balancing loss to reactivate unused vectors, while inter-codebook optimization minimizes redundancy between adjacent quantizers through a structural similarity loss, enhancing expressiveness and overall utilization.

**Adversarial (GAN).** Acoustic tokenizers often apply adversarial losses to improve perceptual quality. A discriminator network $D$ is trained to distinguish between real signals $x$ and reconstructed signals $\hat{x}$, while the tokenizer (generator) is optimized to fool the discriminator. The adversarial loss is defined as a hinge loss over the logits of the discriminator, averaged over multiple discriminators and over time.

The generator loss is:

$$\mathcal{L}_G = \frac{1}{K} \sum_{k=1}^{K} \max(1 - D_k(\hat{x}), 0) \tag{8}$$

The discriminator loss is:

$$\mathcal{L}_D = \frac{1}{K} \sum_{k=1}^{K} \left[ \max(1 - D_k(x), 0) + \max(1 + D_k(\hat{x}), 0) \right] \tag{9}$$

where $D_k(\cdot)$ denotes the output of the $k$-th discriminator. Following VQGAN Esser et al. (2021), the subsequent audio neural vocoders commonly include multiple discriminators with a specific focus on the frequency-level rebuilding to enhance the perceptual quality (Kong et al., 2020; Défossez et al., 2023; Zeghidour et al., 2021). Specifically, these multi-scale discriminators take the stack of multi-resolution or multi-scale complex-valued short-time Fourier transform (STFT) with the real and imaginary parts concatenated as input, e.g., 5 different scales with STFT window lengths of [2048, 1024, 512, 256, 128], for capturing different structures in audio signals.

**Feature Matching (Feat).** To encourage the original and reconstructed signals to exhibit similar abstractions (or to align closely in the latent space), stabilize adversarial training, and encourage more natural reconstructions, a feature-matching loss is often applied. This loss compares intermediate activations from the discriminator for real and reconstructed signals. It is defined as:

$$\mathcal{L}_{\text{Feats}} = \frac{1}{KL} \sum_{k=1}^{K} \sum_{l=1}^{L} \frac{\|D_k^l(x) - D_k^l(\hat{x})\|_1}{\text{mean}(\|D_k^l(x)\|_1)}, \tag{10}$$

where $K$ is the number of discriminators, $L$ is the number of layers in each discriminator, and $D_k^l(\cdot)$ denotes the output of the $l$-th layer of the $k$-th discriminator. Feature matching encourages the generator to match higher-level statistics of real signals, improving stability and perceptual quality.

**Diffusion (Diff).** Diffusion loss is used when the decoder is modeled as a conditional denoising diffusion process. A diffusion model (Rombach et al., 2022) progressively adds noise $\epsilon_t$ to the latent representation $z_t$ during the forward process. A conditioned neural network, parameterized by $\theta$, is trained to predict the noise at each timestep. The training objective minimizes the expected difference between the true noise $\epsilon_t$ and the network prediction $\epsilon_\theta(z_t, t, z_q)$, conditioned on the discrete tokens $z_q$:

$$\mathcal{L}_{\text{diffusion}} = \mathbb{E}_{z_0, t, z_q} \left[ \|\epsilon_t - \epsilon_\theta(z_t, t, z_q)\| \right], \tag{11}$$

where $z_q$ represents the discrete conditioning tokens provided during both training and generation. This approach enables the model to recover acoustic features directly from discrete tokens (Yang et al., 2024e; San Roman et al., 2023; Du et al., 2024b).

**Masked Prediction (MP).** Masked prediction loss is commonly used in tokenizers where the encoder and decoder are trained separately. The encoder is trained to predict masked portions of the input sequence, typically capturing phonetic information rather than reconstructing the full waveform. Following the masked language modeling (MLM) paradigm (Devlin et al., 2019), a portion of the input is randomly masked, and the model is optimized to predict the masked frames from the surrounding context. Formally, given an input sequence of frame-level features $\mathbf{X} = \{x_1, x_2, \ldots, x_T\}$, a binary mask $\mathbf{M} \in \{0,1\}^T$ is applied, where $M_t = 1$ indicates a masked position. The masked input is denoted as $\mathbf{X}^{\text{mask}}$, where $x_t^{\text{mask}} = [\text{MASK}]$ if $M_t = 1$, and $x_t^{\text{mask}} = x_t$ otherwise. The encoder processes this masked input to produce latent representations $\mathbf{Z} = f_e(\mathbf{X}^{\text{mask}})$, and the model is trained to minimize prediction loss.

$$\mathcal{L}_{\text{MP}} = \sum_{t=1}^{T} M_t \cdot \ell(Z_t, x_t)$$

where $f_e$ is the encoder network and $\ell(\cdot, \cdot)$ is the cross-entropy loss. This approach is widely used in speech pretraining models such as HuBERT (Hsu et al., 2021) and WavLM (Chen et al., 2022) and is adopted in several semantic tokenizer designs (Lakhotia et al., 2021; Mousavi et al., 2024b)

### 2.4.3 Auxiliary Components

Beyond the main training objectives, neural audio tokenizers often combine auxiliary components to enhance generalization, improve representation learning, and refine specific features. These auxiliary components fall into three categories: disentanglement, semantic distillation, and supervised semantic tokenization[9].

**Disentanglement.** Disentanglement methods separate different speech attributes into distinct representations, reducing redundancy while allowing independent control over acoustic properties and simplifying downstream tasks. One type of disentanglement in the codec focuses on separating speech and background audio embedding space, enabling better bitrate, entropy control (Yang et al., 2021) or speech enhancement (Omran et al., 2023). Those models normally aim to find a latent space where two ideally orthogonal components $\mathcal{Z}_1$ and $\mathcal{Z}_2$ can be conveniently separated, $\mathcal{F}(x) = \mathcal{Z}_1 \otimes \mathcal{Z}_2$, with $\otimes$ representing straightforward operations such as splitting along the channel (Yang et al., 2021).

Another type separates the conceptual and fundamental components of speech, where each component is typically extracted by its specific encoder. $\mathcal{Z}_k = \mathcal{F}_k(x)$. Early attempts obtained efficient and low-bitrate speech coding through speaker and phoneme disentanglement, utilizing separate training (Polyak et al., 2021) or joint training (Jiang et al., 2023). More recently, TiCodec (Ren et al., 2024b) minimizes token usage by separately quantizing time-invariant global embeddings (e.g., timbre) and time-varying features (e.g., phonetic information). FACodec (Ju et al., 2024) decomposes speech into subspaces such as content, prosody, timbre, and acoustic details through supervised techniques. The timbre extractor in FACodec is optimized with a speaker classification loss, while distinct RVQ modules process other components before supervised decoupling. LSCodec (Guo et al., 2025a) introduces a low-bitrate, speaker-decoupled speech codec using a three-stage training framework with speaker perturbation. A VQ layer is applied after a VAE that disentangles speaker attributes in a continuous space, followed by training a token vocoder on the quantized codes. Unlike most acoustic tokens that redundantly encode speaker timbre across time steps, LSCodec minimizes this inefficiency by isolating timbre from content and prosody. SoCodec (Guo et al., 2024a) employs multi-stream phonetic sequences and ordered product quantization to encode speech into phonetic and time-variant token sequences using HuBERT as a pretrained SSL model. An ECAPA-TDNN-based encoder (Desplanques et al., 2020) extracts an utterance-level global embedding to retain time-invariant information, such as speaker identity, global speaking style, and acoustic environment. SD-Codec (Bie et al., 2025) integrates audio coding with source separation by assigning different audio domains (such as speech, music, and sound effects) to distinct codebooks using multiple parallel RVQ modules.

---

[9]We here inherit the original terminology from the referenced papers (i.e., semantic distillation, and supervised semantic tokenization) . However, it is important to note that both methods typically extract or learn information from SSL features, which predominantly encode phonetic information.

**Semantic Distillation.** Semantic distillation enhances codec representations by incorporating phonetic information into specific codebooks. Various approaches have been explored to distill phonetic knowledge into tokenization while maintaining good reconstruction. Pretrained model guidance is a common approach, where models like SpeechTokenizer (Zhang et al., 2024a), X-Codec (Ye et al., 2025), and Mimi (Défossez et al., 2024) use SSL features to guide specific RVQ layers to learn information from such SSL features. This distillation is implemented by applying regression or classification loss on the first RVQ output to align it with continuous SSL embeddings or discrete SSL tokens. In this way, the first RVQ layers are trained to learn more phonetic information, while later layers focus on acoustic details. Another method injects semantic knowledge directly into the quantizer codebook. LLM-Codec (Yang et al., 2024b) follows this approach by initializing codebooks with token embeddings from LLaMa2 (Touvron et al., 2023) and keeping them frozen during training. This strengthens the ability of the codec to encode meaningful linguistic representations. Some models integrate semantic features into the encoder-quantizer pipeline by combining pretrained SSL representations with acoustic features through concatenation. SemantiCodec (Liu et al., 2024a) and X-Codec (Ye et al., 2025) adopt a dual-encoder-decoder architecture to process SSL semantic tokens independently from acoustic features.

**Supervised Semantic Tokenization.** Some tokenizers explicitly capture phonetic detail through supervised training. For example, Supervised Semantic Speech (S3) (Du et al., 2024a;b) employ a single-codebook VQ layer and FSQ, positioned between two transformer encoder modules. Recently, Har-Tuv et al. (2025) proposed adding phonetic classification auxiliary loss over the first codebook of the RVQ. These models optimize representations using an automatic speech recognition (ASR) loss. Additionally, they utilize optimal-transport conditional flow matching (OT-CFM) (Tong et al., 2024) to model and generate Mel spectrogram distributions conditioned on the produced discrete speech tokens. These supervised approaches produce discrete tokens that effectively preserve phonetic information, making them more aligned with content information and suitable for understanding tasks in speech LMs (Zeng et al., 2024).

### 2.5 Streamability and Domain Categorization

Beyond architecture and training paradigms, audio tokenizers also differ in their support for streaming and their domain of application.

**Streamability.** Streamability refers to the ability of a tokenizer to process and generate audio in real-time with minimal latency, using little or no future context. This property is critical for low-latency applications such as real-time communication or streaming. Latency can be analyzed from two main perspectives:

- **Algorithmic latency**, determined by the look-ahead window—i.e., how much future information is needed to compute the current frame. CNN-based models (Défossez et al., 2023) support streamability via causal convolutions, while Transformer-based (Wu et al., 2024d) models require causal attention mechanisms.

- **Computational complexity**, which becomes especially important when deploying models on resource-constrained systems like mobile or edge devices. Traditional and early neural audio codecs generally maintain low complexity for real-time feasibility. For instance, LPCNet (Valin & Skoglund, 2019b) achieves real-time performance with fewer than 2M parameters at 1.6 kbps. In contrast, more recent models like Encodec scale up to ∼14M parameters to support 1.5 kbps, while BigCodec pushes further to 159M parameters at just 1.04 kbps to improve quality at low bitrates.

Many SSL-based tokenizers rely on non-causal encoders, which limits their use in real-time settings. Thus, achieving streamability with high-quality and efficient causal architectures remains an open research challenge.

**Target Domain.** Some models (Xin et al., 2024; Zhang et al., 2024a; Mousavi et al., 2024b; Défossez et al., 2024) are specifically designed for *speech* tasks such as ASR and TTS. Others are optimized for *music* (Petermann et al., 2021; Tang et al., 2024b) generation and enhancement, capturing tonal and harmonic

Table 2: Audio tokenizers, their characteristics, and abbreviations used throughout the study. As abbreviations, we denote tokenizers as `[name]-[domain(s)]-[sample rate]`.

| Tokenizer | Abbreviations | Domain | | | SR | Frame | #Codes | Params | MACs | Link |
|---|---|---|---|---|---|---|---|---|---|---|
| | | Speech | Music | Audio | (kHz) | Rate | | (Mil) | (G) | |
| EnCodec | Enc-SMA-24 | ✓ | ✓ | ✓ | 24 | 75 | 1024 | 14.9 | 6.1 | Link |
| | Enc-M-32 | | ✓ | | 32 | 50 | 2048 | 56.9 | 14.4 | Link |
| | Enc-A-16 | | | ✓ | 16 | 50 | 2048 | 56.8 | 14.0 | Link |
| DAC | DAC-SMA-44 | ✓ | ✓ | ✓ | 44 | 86 | 1024 | 76.7 | 147.0 | Link |
| | DAC-SMA-24 | ✓ | ✓ | ✓ | 24 | 75 | 1024 | 74.7 | 83.4 | Link |
| | DAC-SMA-16 | ✓ | ✓ | ✓ | 16 | 50 | 1024 | 74.1 | 55.6 | Link |
| SpeechTokenizer | ST-S-16 | ✓ | | | 16 | 50 | 1024 | 103.7 | 17.1 | Link |
| Mimi | Mimi-S-24 | ✓ | | | 24 | 12.5 | 2048 | 79.3 | 8.1 | Link |
| Discrete-WavLM | DWavL-S-16 | ✓ | | | 16 | 50 | 1000 | 331.9 | 21.1 | Link |
| SQ-Codec | SQ-SMA-16 | ✓ | ✓ | ✓ | 16 | 50 | 19683 | 23.5 | 14.7 | Link |
| WavTokenizer | WT-SMA-24 | ✓ | ✓ | ✓ | 24 | 75 | 4096 | 80.6 | 6.3 | Link |
| | WT-S-24 | ✓ | | | 24 | 40 | 4096 | 80.9 | 3.4 | Link |

structures. Some tokenizers (Yang et al., 2024b) are designed for *general audio*, including environmental sounds and non-speech signals. A few models (Ji et al., 2024c; Défossez et al., 2023; Kumar et al., 2023) are trained to handle multiple domains.

## 3 Benchmark Evaluation

Given the wide range of available tokenizers, researchers and practitioners may wonder which *existing* tokenizers are best suited for a given use case. This depends not only on the expected performance for a given task but also on computational efficiency and, in some cases, additional factors such as streamability or the ability to generalize across diverse domains. Several benchmarks have been proposed to evaluate audio tokenizers, offering some guidance on which tokenizers are best suited for different applications and tasks (Wu et al., 2024c; Mousavi et al., 2024a; Shi et al., 2024c; Maimon et al., 2025c).

Nevertheless, drawing solid insights from current benchmarks is challenging, as each focuses on a specific aspect or domain and a holistic comparison of audio tokenizers is missing. Furthermore, while each existing benchmark is internally consistent in its evaluation protocol, they differ significantly in the set of tokenizers they consider, some focus exclusively on acoustic models, while others evaluate semantic tokenizers or even different configurations of the same model (e.g., EnCodec-16k vs. EnCodec-24k). This lack of alignment makes it difficult to derive unified or comparable conclusions across studies. This section contributes to filling this gap by considering a diverse set of publicly available, pre-trained tokenizers across speech, music, and general audio tasks. Unlike previous benchmarks, we perform a joint evaluation across multiple dimensions:

1. *Reconstruction Evaluation and Complexity Analysis.* We assess the quality of resynthesized audio using the original decoder trained for each tokenizer, following protocols from CodecSUPERB and VERSA. We also evaluate the computational efficiency of each tokenizer based on model size (parameters), frame rate, token rate, and multiply-accumulate operations (MACs).

2. *Downstream Evaluation.* We assess the effectiveness of tokenized representations when used directly as input to lightweight models for both discriminative and generative tasks using DASB benchmark.

3. *Acoustic Language Modeling.* We analyze the effectiveness of each tokenizer in training acoustic language models, using the SALMon and Zero-resource benchmarks.

Table 3: Summary of evaluation metrics on resynthesized audio.

| Metric | Functionality | Range | Domain | | |
|--------|---------------|-------|--------|--------|--------|
| | | | Speech | Music | Audio |
| *Signal-level* | | | | | |
| SDR | Signal-to-distortion Ratio | (-inf, inf) | ✓ | ✓ | ✓ |
| SI-SNR | Scale-invariant signal-to-noise ratio | (-inf, inf) | ✓ | ✓ | ✓ |
| PESQ | Perceptual Evaluation of Speech Quality | [1, 5] | ✓ | | |
| UTMOS | UTokyo-SaruLab System for VoiceMOS 2022 | [1, 5] | ✓ | | |
| DNSMOS P808 | Deep Noise Suppression MOS Score of P.808 | [1, 5] | ✓ | | |
| DNSMOS P835 | Deep Noise Suppression MOS Score of P.835 | [1, 5] | ✓ | | |
| PLCMOS | Packet Loss Concealment-focus MOS | [1, 5] | ✓ | | |
| STOI | Short-Time Objective Intelligibility | [0, 1] | ✓ | | |
| VISQOL | Virtual Speech Quality Objective Listener | [1, 5] | | ✓ | ✓ |
| SingMOS | Singing voice MOS | [1, 5] | | ✓ | ✓ |
| *Application-level* | | | | | |
| WER | Word Error Rate (beam=5) | [0, inf) | ✓ | | |
| Spk Sim | Speaker Similarity | [-1, 1] | ✓ | | |

A summary of all tokenizers included in the benchmark evaluation is provided in Table 2. We select these tokenizers based on several factors: (1) We prioritize open-source models with accessible checkpoints and code to ensure reproducibility; (2) we include tokenizers representing a diverse range of quantization strategies—including RVQ (EnCodec, DAC), SVQ (WavTokenizer), FSQ (SQ-Codec), KMeans (Discrete WavLM), and semantically distilled methods (SpeechTokenizer, Mimi); and (3) we aim to cover multiple domains such as speech, music, and general audio, favoring multi-domain tokenizers where available. The impact of single-versus multi-domain tokenizers is further discussed in Section 4. We utilize pre-trained checkpoints released by the original authors. An overview of our benchmark evaluation pipeline is illustrated in Figure 1.

### 3.1 Evaluation for Reconstructed Audio Quality and Complexity

**Background.** We evaluate audio reconstruction quality and examine key properties of audio tokenizers, such as computational complexity, bitrate, and token rate. Reconstruction quality is particularly important for applications like transmission, where preserving signal fidelity is crucial. Moreover, high reconstruction quality might be a useful proxy for selecting effective tokenizers for downstream tasks, especially those that rely directly on the decoder for audio generation, such as speech enhancement and source separation. In such cases, the reconstruction performance of the tokenizer can impact the overall task performance.

**Experimental Setup.** The input audio is first compressed using an audio tokenizer and then resynthesized through its corresponding decoder. The resynthesized audio is assessed from both signal-level and application-level perspectives, providing insights into how well each tokenizer preserves information. To ensure a comprehensive analysis, we integrate the evaluation scripts from Codec-SUPERB and VERSA (Wu et al., 2024c;b; Shi et al., 2024c; 2025). We extend the evaluation across three domains (music, general audio, and speech) to examine how different tokenizers perform in diverse acoustic scenarios.

**Dataset.** For speech evaluation, we use the LibriSpeech test-clean set (Panayotov et al., 2015). For music, we use the MUSDB dataset (Rafii et al., 2017), which consists of approximately 10 hours of full-length and professionally-recorded musical tracks at 44.1kHz. Lastly, for general audio we opt for the Audioset (Gemmeke et al., 2017) test-set, which accounts for approximately 55 hours of audio clips extracted from YouTube.

Table 4: Reconstruction performance of audio tokenizers (speech).

| Tokenizer | #Q | kbps | Token rate | SDR ↑ | SI-SNR↑ | PESQ ↑ | UTMOS ↑ | DNSMOS P808↑ | DNSMOS P835↑ | PLCMOS ↑ | STOI ↑ | WER ↓ | Spk Sim↑ |
|---|---|---|---|---|---|---|---|---|---|---|---|---|---|
| Ground truth | - | - | - | **290.16** | **55.92** | **4.64** | **4.09** | **3.84** | 3.18 | 4.16 | **1.00** | 2.83 | **1.00** |
| Enc-SMA-24 | 2 | 1.5 | 150 | 0.82 | −1.53 | 1.56 | 1.58 | 3.21 | 2.39 | 3.44 | 0.85 | 5.44 | 0.42 |
|  | 8 | 6 | 600 | 6.50 | 4.83 | 2.77 | 3.09 | 3.57 | 2.96 | 4.08 | 0.94 | 2.78 | 0.72 |
|  | 32 | 24 | 2400 | **9.75** | **7.90** | 3.71 | 3.74 | 3.74 | 3.19 | 4.29 | 0.97 | 2.77 | 0.78 |
| DAC-SMA-24 | 2 | 1.5 | 150 | -0.57 | −8.40 | 1.48 | 1.68 | 3.24 | 2.61 | 3.27 | 0.83 | 9.59 | 0.45 |
|  | 8 | 6 | 600 | 1.79 | −9.51 | 3.40 | 3.60 | 3.69 | 3.16 | 4.15 | 0.95 | 3.53 | 0.73 |
|  | 32 | 24 | 2400 | 2.20 | −9.47 | **4.45** | **4.05** | 3.78 | 3.20 | 4.40 | **0.99** | 2.72 | 0.80 |
| ST-S-16 | 2 | 1 | 100 | -7.10 | −14.46 | 1.21 | 2.32 | 3.37 | 2.78 | 2.96 | 0.77 | 4.20 | 0.35 |
|  | 8 | 4 | 400 | 3.01 | 0.53 | 2.62 | 3.84 | 3.77 | 3.17 | 4.00 | 0.92 | 2.41 | 0.86 |
| Mimi-S-24 | 8 | 1.1 | 100 | 3.43 | 1.19 | 2.22 | 3.60 | 3.68 | 3.17 | 4.27 | 0.90 | 3.72 | 0.70 |
|  | 32 | 4.4 | 400 | 9.32 | 7.45 | 3.38 | 3.92 | 3.74 | 3.18 | 4.40 | 0.96 | 2.96 | 0.85 |
| DWavL-S-16 | 2 | 1 | 100 | -13.96 | −37.23 | 1.13 | 3.32 | 3.68 | 3.13 | 3.86 | 0.75 | 4.97 | 0.33 |
|  | 6 | 3 | 300 | -12.69 | −35.43 | 1.19 | 3.32 | 3.72 | 3.13 | 4.05 | 0.75 | 4.34 | 0.35 |
| SQ-SMA-16 | 4 | 3 | 200 | 1.91 | −8.61 | 3.31 | 3.90 | **3.83** | **3.28** | 4.13 | 0.96 | **2.37** | **0.87** |
| WT-SMA-24 | 1 | .98 | 75 | 2.02 | −0.79 | 1.88 | 3.77 | 3.76 | 3.18 | **4.41** | 0.87 | 8.10 | 0.60 |
| WT-S-24 | 1 | .52 | 40 | 0.17 | −3.16 | 2.05 | 3.89 | 3.82 | 3.27 | 4.38 | 0.89 | 8.91 | 0.61 |

**Evaluation Setup.** Table 3 reports the reconstruction metrics for resynthesized audio. Beyond quality, it is important to jointly consider factors often overlooked in the literature, such as computational efficiency and tokenizer properties. To address this, we also include these metrics as Table 2: (1). *Model parameters (Params)*: The total number of parameters in the audio tokenizer. (2). *Computational complexity (MACs)*: Number of arithmetic operations performed by a tokenizer. MACs are computed for a one-second audio sample using PyFlops[10]. For components incompatible with PyFlops, such as streaming self-attention, dot product, calculations are performed manually. (3). *Bitrate*: The number of bits per second, representing a balance between audio quality and compression efficiency. (4). *Frame rate*: The number of temporal frames used to encode one second of audio. (5). *Token rate*: Number of tokens required to encode one second of audio, an important factor for acoustic language modeling applications.

**Results and Discussion.**

**Speech.** From Table 4, we make the following observations: (1) For both EnCodec and DAC, the reconstruction quality consistently degrades as the bitrate decreases from 24k to 6k and 1.5k. This trend confirms that higher bitrates better preserve acoustic detail, resulting in improved reconstruction quality across all evaluated metrics. (2) For SpeechTokenizer (4k vs. 1k) and Mimi (4.4k vs. 1.1k), which both apply semantic distillation to the first codebook, all objective metrics decline at lower bitrates. However, the WER does not drop as drastically, indicating that semantic distillation effectively preserves linguistic content even when the overall reconstruction quality decreases. (3) Discrete WavLM exhibits significantly lower SDR, SI-SNR, PESQ, STOI, and Spk-Sim scores. Since these metrics rely on reference ground truth signals, the poor performance indicates these models are not optimized for precise waveform reconstruction. Metrics such as UTMOS, DNSMOS, and PLCMOS, however, remain reasonable, suggesting these tokenizers still preserve speech quality. This discrepancy indicates that discrete tokenizers focus more on high-level representations than on exact waveform reconstruction. (4) SQ-SMA-16 performs comparable or even better than large bitrate codec models (e.g., Mimi-S-24 4.4kbps, and DAC-SMA-24 6kbps). (5) Finally, we find that SDR and SI-SNR are less reliable indicators. A possible reason is that the signal is over-compressed, the generation of neural codec (especially in low-bitrate), usually have less consistency in the local sample-level information. It is likely due to non-linear shifts or amplitude variations.

---

[10] https://github.com/sovrasov/flops-counter.pytorch/tree/master

Table 5: Reconstruction performance of audio tokenizers for both general audio and music experiments.

| Tokenizer | #Q | kbps | Token rate | Audio | | | | | Music | | | | |
|---|---|---|---|---|---|---|---|---|---|---|---|---|---|
| | | | | SDR↑ | CI-SDR↑ | SI-SNR↑ | VISQOL↑ | Sing MOS↑ | SDR↑ | CI-SDR↑ | SI-SNR↑ | VISQOL↑ | Sing MOS↑ |
| Ground truth | - | - | - | **252.75** | **84.90** | **57.96** | **4.73** | **2.70** | **254.24** | **87.26** | **60.26** | **4.73** | **2.79** |
| Enc-SMA-24 | 2 | 1.5 | 150 | −1.29 | −1.28 | −4.31 | 3.94 | 2.59 | 2.16 | 2.13 | 0.46 | 4.05 | 2.67 |
| | 8 | 6 | 600 | 4.28 | 4.10 | 2.33 | 4.25 | 2.60 | 7.32 | 7.17 | 5.87 | 4.38 | 2.66 |
| | 32 | 24 | 2400 | **7.72** | **7.33** | 5.64 | 4.36 | 2.60 | **11.04** | **10.75** | **9.19** | 4.50 | 2.66 |
| DAC-SMA-24 | 2 | 1.5 | 150 | −2.60 | −2.55 | −11.55 | 3.99 | 2.59 | 1.75 | 1.71 | −2.21 | 3.94 | **2.70** |
| | 8 | 6 | 600 | 1.35 | 1.22 | −10.28 | 4.35 | 2.61 | 4.82 | 4.67 | −1.25 | 4.30 | 2.68 |
| | 32 | 24 | 2400 | 2.45 | 2.22 | −9.91 | **4.59** | 2.60 | 5.56 | 5.37 | −1.16 | **4.56** | 2.66 |
| SQ-SMA-16 | 4 | 3 | 200 | −2.33 | −2.33 | −10.50 | 4.32 | **2.62** | 3.44 | 3.39 | −0.38 | 4.34 | 2.68 |
| WT-SMA-24 | 1 | .98 | 75 | −4.55 | −4.45 | **9.78** | 3.96 | 2.56 | −14.30 | −14.28 | −23.09 | 3.64 | 2.60 |
| WT-S-24 | 1 | .52 | 40 | −11.00 | −10.85 | −20.91 | 3.85 | 2.53 | −19.91 | −19.89 | −45.55 | 3.33 | 2.42 |

**General Audio and Music.** Table 5 summarizes the reconstruction results for the general audio and music domains. As in the speech domain, reconstruction quality generally decreases with lower bitrates. The results also reveal notable trends related to both the training domain and optimization objectives for each tokenizer. EnCodec achieves the best overall reconstruction performance across SDR, SI-SNR, and perceptual metrics (VISQOL and SingMOS), particularly at higher bitrates. In contrast, DAC shows surprisingly poor performance in time-domain metrics, with negative SI-SNR values in most settings. This suggests that DAC relies on adversarial or perceptual optimization strategies that do not prioritize time-domain reconstruction loss. Nonetheless, despite poor time-domain fidelity, DAC maintains strong VISQOL and SingMOS scores, indicating its reconstructions remain perceptually plausible.

Similar to DAC, WavTokenizer is not explicitly optimized with a time-domain waveform reconstruction loss, resulting in poor SDR and SI-SNR scores. However, its perceptual metrics (VISQOL and SingMOS) remain relatively strong. This further highlights the limitations of time-domain metrics in evaluating token-based representations, as previously observed in the speech reconstruction results. Among the two WavTokenizer variants, WT-S-24 and WT-SMA-24, only WT-S-24 is fully out-of-domain, having been trained exclusively on speech data. Both models exhibit poor performance across objective metrics such as SDR and SI-SNR, as well as perceptual metrics (VISQOL and SingMOS). This degradation stems from a combination of factors: the domain mismatch in WT-S-24, the use of the lowest bitrate among all models, and the absence of explicit waveform reconstruction losses during training.

**Summary.** Overall, these results underscore the importance of evaluating audio tokenizers beyond traditional waveform fidelity measures. Models optimized for perceptual or downstream tasks may exhibit low signal reconstruction performance, yet still produce subjectively high-quality audio reconstructions.

### 3.2 Downstream Evaluation

**Background.** Evaluating token quality solely based on reconstruction performance raises an important question: *how much task-relevant information is preserved in the tokens, independent of the decoder's capacity?* This distinction is critical in multimodal language modeling settings, where audio tokens are used directly as input to large language models. These models must perform both discriminative tasks (e.g., ASR, emotion recognition) that map audio to text, and generative tasks (e.g., speech synthesis, speech-to-speech translation) that output audio.

**Experimental Setup.** We evaluate discrete audio tokenizers using the DASB benchmark (Mousavi et al., 2024a), built on the SpeechBrain toolkit (Ravanelli et al., 2024), which isolates the representational quality of the tokens for downstream modeling. For each task, the encoder is frozen, and a task-specific classification

Table 6: Datasets, metrics, and downstream models for the DASB evaluation.

| Task | Dataset | Architecture | Metric(s) | Data Link |
|------|---------|--------------|-----------|-----------|
| *Speech (Discriminative)* | | | | |
| ASR (En) | LibriSpeech (Korvas et al., 2014) | Branchformer | WER | Link |
| ASR (Low-resource) | CommonVoice 17.0 (Ardila et al., 2020) | BiLSTM | WER | Link |
| Speaker ID / Verification | VoxCeleb1 (Nagrani et al., 2017) | ECAPA-TDNN | Accuracy / EER | Link |
| Emotion Recognition | IEMOCAP (Busso et al., 2008) | ECAPA-TDNN | Accuracy | Link |
| Keyword Spotting | Speech Commands (Warden, 2018) | ECAPA-TDNN | Accuracy | Link |
| Intent Classification | SLURP (Bastianelli et al., 2020) | BiLSTM+Linear | Accuracy | Link |
| *Speech (Generative)* | | | | |
| Speech Enhancement | VoiceBank (Valentini-Botinhao et al., 2016) | Conformer | DNSMOS / dWER | Link |
| Speech Separation | Libri2Mix (Cosentino et al., 2020) | Conformer | DNSMOS / dWER / SpkSim | Link |
| *Music* | | | | |
| Music Genre Classification | GTZAN (Tzanetakis & Cook, 2002) | ECAPA-TDNN | Accuracy | Link |
| Music Source Separation | MUSDB (Rafii et al., 2017) | Conformer | SDR / SIR / SAR | Link |
| *General Audio* | | | | |
| Sound Event Classification | ESC-50 (Piczak, 2015) | ECAPA-TDNN | Accuracy | Link |
| Audio Separation | FUSS (Wisdom et al., 2021) | Conformer | SDR | Link |

head is trained. We use lightweight classification heads to avoid hiding weaknesses in the token representations. Generative tasks additionally use the frozen decoder. All token embeddings are projected to a fixed dimensionality of 1024 to ensure consistency across models. This value corresponds to the largest embedding size among the tokenizers in our benchmark. For tokenizers with multiple codebooks, a weighted sum of codebook embeddings is computed, with weights learned jointly with the downstream head (Chen et al., 2022; Zaiem et al., 2023). For SQ-Codec, which uses scalar quantization and group vector quantization, we apply a ternary matrix-based embedding and concatenate four 256-dimensional group vectors to match the 1024-dimensional standard. Each tokenizer is evaluated across multiple bitrate settings (low, high, and recommended). We tune the most relevant hyperparameters, such as learning rate and model capacity, using the Tree-structured Parzen Estimator (TPE) (Bouthillier et al., 2022) with 20 trials. To obtain a more robust performance estimate, we average the results of each tokenizer over three downstream training runs with different random seeds. For ASR tasks, both character-level and byte pair encoding (BPE) segmentations are considered, and the better-performing configuration is reported. Table 6 summarizes the benchmark tasks and their corresponding downstream models. When multiple-domain tokenizer checkpoints are available, we use the multi-domain version for consistency. The impact of domain-specific vs. multi-domain training is further analyzed in our ablation study (Section 4). Additional implementation details are provided in the DASB paper (Mousavi et al., 2024a).

**Dataset.** We evaluate audio tokenizers across diverse tasks and domains, including speech discriminative tasks such as ASR, low-resource ASR (L-R ASR), speaker identification and verification (SID, SV), emotion recognition (ER), intent classification (IC), and keyword spotting (KS). For generative tasks, we include speech enhancement (SE) and speech separation (SS). In the music and general audio domains, we evaluate music genre classification (MG), music source separation (MSS), general sound separation (ASS), and sound event classification (SEC). A full summary of datasets and tasks is provided in Table 6.

**Evaluation Setup** For continuous baselines, we follow Zaiem et al. (2023) by using a weighted sum of WavLM-large layers as input across most tasks. To ensure a fair comparison between discrete and continuous representations, we adopt identical downstream architectures for both settings. While neither WavLM nor the chosen downstream architecture may represent the state-of-the-art for every task, using a consistent setup across all experiments allows us to isolate the effect of representation quality. For instance, in speech separation, well-established baselines such as Conv-TasNet (Luo & Mesgarani, 2019) and Transformer-based

models (Saijo et al., 2024) are excluded. Libri2Mix has become a saturated benchmark, with many approaches reaching near-ceiling performance, and adding stronger backbones would not yield meaningful insights. Instead, our focus is on isolating the effects of discrete versus continuous representations under a shared architecture. There are two exceptions: for music and general audio separation, tasks that remain more challenging, we use stronger, task-specific architectures for the continuous baselines. Specifically, we adopt DEMUCS (Rouard et al., 2023) for music source separation and TDCN++ (Kavalerov et al., 2019) for general sound separation, as these models are better suited to the complexity of these domains. We evaluate each task using standard, task-specific metrics (Table 6): ASR is evaluated using Word Error Rate (WER); SV uses Equal Error Rate (EER); classification tasks including ER, SID, IC, MG, and SEC are evaluated using classification accuracy (ACC). For SE and SS, we report DNSMOS (Reddy et al., 2022) for perceptual audio quality, differential WER (dWER) using Whisper (Radford et al., 2023) for intelligibility, and speaker similarity (SpkSim) based on cosine similarity of WavLM-derived embeddings. Music source separation is evaluated using signal-to-distortion ratio (SDR), signal-to-artifact ratio (SAR), and signal-to-interference ratio (SIR) via the BSSEval toolkit (Vincent et al., 2006), while general sound separation is assessed using SDR on FUSS (Wisdom et al., 2021). All results are averaged over three runs with different random seeds to ensure robustness.

**Results and Discussion**    Tables 7, 8, and 9 summarize performance across discriminative and generative tasks for speech, music, and general audio. Below, we outline key findings from our experiments.

**Speech Tasks.** Discrete WavLM consistently performs best in discriminative tasks, likely due to its strong ability to preserve phonetic content. SpeechTokenizer, which uses semantic distillation, ranks second. In speaker recognition, however, DAC achieves the best results, suggesting that reconstruction-based objectives help preserve speaker identity. For speech separation and enhancement, WavLM performs well at low and medium bitrates but shows poor results in speaker similarity metrics. This aligns with previous findings (van Niekerk et al., 2022) that SSL-based tokenizers tend to lose speaker-related information. Another notable observation is that in many cases, the reconstructed DNSMOS score, representing the upper bound set by the codec alone without any separation, does not surpass the score obtained by using the raw mixture as the estimate (i.e., the lower bound), suggesting that limitations in reconstruction quality may constrain downstream performance, particularly for high-fidelity tasks like speech separation.

**Audio and Music Tasks.** For general audio and music tasks, EnCodec consistently outperforms other tokenizers across all bitrates and domains, while DAC lags behind. Although DAC is known for strong perceptual quality, its signal-level fidelity is generally lower, which likely impacts its separation performance. The difficulty of these tasks is evident from the SI-SDR of the unprocessed mixtures, for example, approximately -16 dB for general audio and -7.7 dB for music. Even the best-performing model (EnCodec at medium bitrate) only reaches about -7 dB SI-SDR for audio and -5.7 dB for music. We report the performance in terms of SI-SDR improvement ("SI-SDRi"), not absolute SI-SDR. Thus, the reason we also provide the performance on "unprocessed" mixtures. For instance, for general audio, we report an improvement of 9.53 dB over the mixture (-16.5 dB). In absolute value, this means that the resulting predictions yield an average of -7-7 dB performance. While high-bitrate settings have proven to be challenging for downstream tasks, they perform particularly poorly in music separation, emphasizing that increasing bitrate alone does not improve separation quality and may even degrade performance. This may be due to the inherently polyphonic and less sparse nature of music (in contrast to speech and general audio), which results in highly overlapping sources that are harder to disentangle from detailed but semantically entangled representations.

**Impact of Codebook Size.** Increasing the number of codebooks (e.g., 2, 8, 32) improves signal reconstruction but often reduces downstream task performance. This trade-off suggests that while more codebooks enhance fidelity, they often degrade performance for both discriminative and generative tasks by increasing output dimensionality and modeling complexity. In RVQ-based models, earlier codebooks capture more phonetic information, while later ones often add redundancy, which may explain this trade-off. This highlights an important design principle for tokenizers: *optimizing for reconstruction alone does not guarantee better performance on downstream tasks.* Medium bitrate settings typically provide the best balance between audio reconstruction quality and task performance.

**Discrete vs Continuous.** While discrete tokens show promise, they face notable limitations in complex scenarios such as polyphonic music separation or noisy environments. Continuous features consistently outperform discrete tokens due to the information loss inherent in quantization, which affects critical attributes like phonetics, emotion, and speaker identity. These limitations are further exacerbated in low-resource settings. For instance, although Discrete WavLM performs competitively at low and medium bitrates for low-resource ASR, it still lags behind the continuous baseline. RVQ-based tokenizers struggle even more, especially on smaller datasets such as Welsh, ESC-50, and GTZAN, with high bitrate amplifying these issues. Performance improves with more data: Discrete WavLM, for example, achieves 6.0% WER on LibriSpeech (960h), 22.0% on Basque (116h), and 58.9% on Welsh (8h) at low bitrate using a BiLSTM head, illustrating a strong correlation between data scale and ASR accuracy. From the hyperparameter tuning experiments (not reported here for brevity), we noticed that larger downstream models help improve convergence and performance, particularly for acoustic tokenizers, which are more sensitive to both data scale and model capacity. Semantic tokenizers are generally more robust in low-resource settings but still fall short of continuous representations with extremely limited data. Overall, careful tuning and appropriate scaling of both data and model are essential for an effective use of discrete representations, especially acoustic tokens.

**Summary.** Semantic tokenizers (e.g., Discrete WavLM) are generally more robust, especially in low-resource settings, but still fall short of continuous representations when data is limited. Training downstream models with semantic (Discrete WavLM) or semantically distilled tokenizers (Mimi and SpeechTokenizer) tends to be more stable and reliable compared to acoustic tokenizers (EnCodec, DAC, WavTokenizer, and SQ-Codec), which often require larger datasets and more careful model scaling. Overall, discrete tokenizers are more sensitive to architectural choices and hyperparameters of the downstream head, whereas continuous features typically yield more consistent performance across configurations. Therefore, careful tuning and appropriate scaling of both data and model architecture are crucial for effectively leveraging discrete representations. While discrete tokens offer advantages in efficiency and modularity, continuous representations still lead in overall performance. Bridging this gap is essential for the successful integration of audio tokens into future multimodal language models.

Table 7: DASB results for discriminative tasks (speech).

| Tokenizer | #Q | ASR-En | | ASR-LR | | ER | IC | KS | SI | SV |
|---|---|---|---|---|---|---|---|---|---|---|
| | | WER↓ | | WER↓ | | ACC↑ | ACC↑ | ACC↑ | ACC↑ | EER↓ |
| | | Clean | Other | Welsh | Basque | | | | | |
| Continuous | – | **4.07** | **6.81** | **41.77** | **14.32** | **63.10** | **86.10** | **99.00** | **99.70** | **2.10** |
| Enc-SMA-24 | 2 | 12.70±0.37 | 29.09±0.13 | 90.90±0.32 | 51.00±0.98 | 45.50±0.02 | 42.90±0.16 | 77.73±3.12 | 89.81±5.46 | 18.33±0.26 |
| | 8 | 8.43±0.13 | 21.77± 0.36 | 84.53±1.90 | 45.36±0.57 | 44.73±0.02 | 40.03±0.29 | 74.30±1.69 | 94.26±3.99 | 13.54±0.57 |
| | 32 | 9.95±1.17 | 23.24± 1.22 | 97.39±1.19 | 58.21±0.92 | 42.96±0.02 | 33.66±2.65 | 69.10±3.42 | 91.12±1.92 | 10.12±6.66 |
| DAC-SMA-24 | 2 | 14.84±0.25 | 33.88±0.20 | 95.21±0.84 | 68.93±0.42 | 45.20±0.01 | 29.83±0.19 | 67.27±1.56 | **97.88±0.79** | 21.80±1.00 |
| | 8 | 10.73± 0.10 | 25.39± 0.20 | 97.20±0.14 | 62.45±1.40 | 44.73±0.02 | 23.97±0.41 | 65.27±2.82 | 87.33±10.98 | 15.86±5.26 |
| | 32 | 13.13±0.16 | 28.47±0.19 | 98.96±0.18 | 73.57±1.56 | 43.20±0.02 | 44.60±39.19 | 68.67±2.91 | 87.69±4.99 | 17.12 ± 0.76 |
| ST-S-16 | 2 | 9.48±0.10 | 22.68±0.10 | 71.36±0.32 | 42.17±0.05 | 54.86±0.01 | 56.80±0.08 | 94.11±0.63 | 73.16±0.37 | 24.23±0.29 |
| | 8 | 9.06± 0.45 | 21.72±0.23 | 68.36±0.44 | 35.35±0.22 | 55.00±0.01 | 53.83±0.05 | 94.11±0.07 | 96.78±0.45 | 10.45±0.43 |
| Mimi-S-24 | 8 | 9.73±0.61 | 22.65±0.41 | 91.59±0.15 | 59.18±8.52 | 51.13±0.02 | 53.83±0.19 | 92.18±0.20 | 79.50±0.43 | 18.68±0.35 |
| | 32 | 10.84±0.56 | 24.10±0.36 | 96.89±0.07 | 58.15±6.90 | 46.76±0.01 | 50.73±0.50 | 91.31±0.19 | 63.93±13.64 | 23.91±4.60 |
| DWavL-S-16 | 2 | **4.78±0.25** | 10.58±0.17 | 58.98±0.15 | 22.02±0.17 | 61.53±0.02 | 76.33±0.17 | **96.82±0.92** | 76.57±0.33 | 22.41±0.19 |
| | 6 | 5.07±0.17 | **9.57±0.20** | **48.94±0.38** | **19.66±0.33** | **63.20±0.01** | **78.73±0.12** | 95.89±0.50 | 92.31±0.09 | 13.47±0.22 |
| SQ-SMA-16 | 4 | 91.57±0.49 | 92.90±0.41 | 94.80±0.88 | 94.24±1.24 | 41.30±0.06 | 58.13±0.26 | 92.74±0.42 | 97.38±0.03 | **9.69±0.25** |
| SQ-SMA-16* | 4 | 11.63±0.08 | 30.91±0.17 | – | – | – | – | – | – | – |
| WT-SMA-24 | 1 | 16.11±0.18 | 35.48±0.35 | 97.41±0.08 | 75.82±0.20 | 43.43±0.02 | 15.25±0.15 | 59.13±2.10 | 85.90±2.48 | 19.38±0.36 |

## 3.3 Acoustic Language Models Evaluation

Following the rise of LLMs, researchers have extended the generative auto-regressive framework beyond text to discrete representations of speech (Lakhotia et al., 2021), audio (Borsos et al., 2023b), and music (Copet

Table 8: DASB results for generative tasks (speech).

| Models\Tasks | #Q | SE | | | SS – Speech | | | |
|---|---|---|---|---|---|---|---|---|
| | | DNSMOS↑ | dWER↓ | Spk Sim↑ | DNSMOS Rec↑ | DNSMOS Sep↑ | dWER↓ | Spk Sim↑ |
| Continuous | – | 3.49 | **4.92** | **0.93** | – | 3.68 | **9.97** | **0.94** |
| Enc-SMA-24 | 2 | 3.15±0.01 | 34.95±0.64 | 0.86±0.00 | 3.19 | 3.13±0.00 | 80.33±1.77 | 0.88±0.00 |
| | 8 | 3.08±0.01 | 22.70±1.84 | 0.88±0.00 | 3.54 | 3.08±0.00 | 53.37±0.65 | 0.90±0.00 |
| | 32 | 2.78±0.01 | 65.70±6.09 | 0.80±0.01 | 3.72 | 2.97±0.01 | 92.42±0.97 | 0.85±0.00 |
| DAC-SMA-24 | 2 | 3.26±0.01 | 54.85±1.82 | 0.86±0.00 | 3.16 | 3.01±0.00 | 101.19±1.99 | 0.85±0.00 |
| | 8 | 3.51±0.01 | 29.44±3.93 | **0.90±0.01** | 3.67 | 3.30±0.00 | 52.77±2.48 | **0.93±0.00** |
| | 32 | 2.93±0.01 | 30.66±0.97 | 0.88±0.00 | 3.76 | 2.67±0.01 | 92.07±0.05 | 0.88±0.01 |
| ST-S-16 | 2 | 3.19±0.02 | 29.98±0.58 | 0.86±0.00 | 3.20 | 3.13±0.00 | 84.94±0.63 | 0.87±0.00 |
| | 8 | 3.49±0.01 | 21.65±0.57 | 0.87±0.00 | 3.72 | 3.43±0.01 | 60.90±0.77 | 0.91±0.00 |
| Mimi-S-24 | 8 | 3.25±0.01 | 67.56±2.21 | 0.85±0.00 | 3.65 | 3.29±0.00 | 109.30±3.30 | 0.87±0.00 |
| | 32 | 3.18±0.01 | 102.61±2.40 | 0.82±0.00 | 3.72 | 3.00±0.00 | 137.00±2.16 | 0.82±0.00 |
| DWavL-S-16 | 2 | 3.56±0.01 | 25.88±2.15 | 0.88±0.00 | 3.57 | 3.56±0.00 | 49.57±0.64 | 0.85±0.00 |
| | 6 | **3.57±0.01** | **9.43±0.33** | 0.89±0.00 | 3.75 | **3.75±0.01** | **30.39±0.45** | 0.91±0.00 |
| SQ-SMA-16 | 4 | 3.28±0.01 | 122.33±8.74 | 0.83±0.00 | **3.77** | 3.19±0.00 | 136.00±3.58 | 0.83±0.00 |
| WT-SMA-24 | 1 | 3.33±0.01 | 67.53±10.65 | 0.85±0.00 | 3.57 | 3.42±0.00 | 118.33±4.50 | 0.86±0.00 |
| Mixture | – | – | – | – | – | 3.43 | – | – |

Table 9: DASB results for generative and discriminative tasks (music and general audio).

| Tokenizer | #Q | SS – Audio | | SS – Music | | | | SEC | MGC |
|---|---|---|---|---|---|---|---|---|---|
| | | SI-SDRi↑ | | SI-SDRi↑ | | SAR↑ | SIR↑ | ACC↑ | ACC↑ |
| | | Rec | Sep | Rec | Sep | | | | |
| Continuous | – | – | **15.07** | – | **13.29** | **9.56** | **11.99** | **92.91** | **87.00** |
| Enc-SMA-24 | 2 | 0.76 | 7.03±0.49 | 3.36 | 1.49±2.04 | -2.80±1.68 | **5.96±1.52** | 34.83±0.47 | **70.33±1.70** |
| | 8 | 3.87 | **9.53±0.33** | 7.99 | **1.98±0.36** | **-1.95±0.33** | 5.26±0.22 | **37.00±0.73** | 54.67±3.86 |
| | 32 | **5.76** | -1.73±0.09 | **11.10** | -11.72±0.35 | -15.00±0.02 | -0.42±0.01 | 35.43±1.45 | 39.67±1.25 |
| DAC-SMA-24 | 2 | 0.12 | 3.84±0.48 | 2.37 | 1.01±0.17 | -3.59±0.09 | **5.92±0.28** | 31.03±1.84 | 50.00±0.82 |
| | 8 | 3.33 | 5.62±0.21 | 6.66 | -11.77±0.1 | -10.62±2.35 | -5.52±3.68 | 28.60±0.79 | 47.67±3.09 |
| | 32 | 4.73 | -4.92±0.32 | 8.54 | -11.32±0.12 | -12.70±0.17 | -2.05±0.41 | 36.67±0.92 | 50.00±0.82 |
| SQ-SMA-16 | 4 | 3.62 | 6.54±0.22 | 5.53 | -3.62±0.87 | -5.84±0.86 | 1.42±0.32 | 31.37±1.37 | 42.67±0.47 |
| WT-SMA-24 | 1 | -24.05 | -16.72±0.08 | -2.66 | -4.52±0.04 | -8.32±0.07 | 2.65±0.11 | 34.50±0.82 | 48.00±1.41 |
| Mixture | – | – | -16.5 | – | -7.71 | 50.01 | -inf | – | – |

et al., 2023). *Acoustic language models* refer to models that learn to represent or generate audio signals (including speech, general audio, and music) using this generative autoregressive framework. This modeling approach has proven highly effective across domains, enabling powerful generation capabilities (Défossez et al., 2024). In this section, we begin by examining unconditional speech generation (Speech Language Models) and text-conditioned generation (text-to-speech (TTS)). We then explore audio generation and finally turn to the music modality.

### 3.3.1 Speech Language Modeling

**Background.** Speech Language Models (SLMs) have gained significant interest (Arora et al., 2025; Wu et al., 2024a; Peng et al., 2024; Cui et al., 2024; Ji et al., 2024a; Latif et al., 2023), demonstrating remarkable performance in traditional speech tasks (Chen et al., 2025a; Elmakies et al., 2025), diverse generative applications (Yang et al., 2024c;b), and reasoning over speech and audio signals (Chu et al., 2023; Wang et al., 2025a; Yosha et al., 2025).

SLMs can generally be classified into two main categories: (i) generative SLMs that are conditioned on previous speech/text tokens and generate speech/text (Défossez et al., 2024; Cuervo et al., 2025; Nguyen et al., 2025), and (ii) speech-aware LMs that are conditioned on speech/text and generate text (Chu et al., 2023; Tang et al., 2024a; Mousavi et al., 2025). This work focuses on the first category of SLMs as there is a growing interest from the research community to study generative SLMs (Nguyen et al., 2025; Maimon et al., 2025b; Rubenstein et al., 2023).

Several SLMs operate over discrete speech representations derived from a pre-trained SSL model (Nguyen et al., 2025; Lakhotia et al., 2021; Cuervo et al., 2025). Others employ semantically distilled acoustic tokenizers (Défossez et al., 2024) or adopt a cascading, mixed-resolution strategy (Borsos et al., 2023b), modeling speech hierarchically from coarse semantic content to fine acoustic details, using language models conditioned on previously generated streams. More recently, supervised semantic tokenizers have gained popularity. These methods typically quantize the output layer of a pre-trained ASR system to produce discrete tokens (Zeng et al., 2024; 2025). In this study, we analyze the impact of these choices by comparing SLMs trained under a controlled setup with different audio tokenizers presented in Table 2.

We focus on SLMs that model the joint probability of a sequence of speech tokens $\mathbf{q}$ as:

$$P(\mathbf{q} = q_1, \ldots, q_n) = \prod_{i=1}^{n} \mathbb{P}(q_i \mid q_{<i}), \tag{12}$$

where $q_i \in \mathcal{V}_q$, and $\mathcal{V}_q$ denotes the vocabulary of speech tokens. These models are typically implemented as decoder-only transformers (Vaswani et al., 2017) and trained to minimize the negative log-likelihood:

$$\mathcal{L}_{LM} = -\sum_{i=1}^{n} \mathbb{P}(q_i | q_{<i}). \tag{13}$$

Each token is embedded via a matrix $E \in \mathbb{R}^{|\mathcal{V}_s| \times d}$, where $d$ denotes the embedding dimension. The resulting sequence is processed by a stack of causal transformer layers, yielding contextual representations $\mathbf{c} \in \mathbb{R}^{n \times d}$. A final linear projection $U \in \mathbb{R}^{d \times |\mathcal{V}_s|}$ maps these to logits, which are converted to a probability distribution over the vocabulary via a softmax: $p(q_{i+1} \mid \mathbf{c}_i)$.

**Experimental Setup.** Motivated by Maimon et al. (2025a), each SLM is built upon the Qwen-2.5 architecture (Yang et al., 2024a) (357M parameters in total, after removing the text embedding tables) and initialized using TWIST (Hassid et al., 2023). The textual embedding tables are replaced with new audio embedding tables corresponding to the audio codebooks. The models are trained using the standard cross-entropy loss reported in Eq. 13. To ensure all SLMs processed a comparable amount of audio during training, we dynamically adjusted the tokens per batch according to their tokenizer's frame rate. To accommodate multiple codebooks, the delay pattern from MusicGen (Copet et al., 2023) is applied across all SLM models. The models are trained for a total of 50,000 optimizer steps, with a context length set to 1024. The audio target batch size is set to include about 2.9 hours of speech per backpropagation step. We used the Adam optimizer coupled with a linear learning rate scheduler, applying a 1% warmup ratio (corresponding to 500 steps). All input samples are fed to the SLMs using a packing strategy by concatenating all samples together until having a sequence of the target length. To ensure fairer evaluations across different tokenizers, the number of codebooks is restricted to a maximum of 8, promoting better alignment between them. All code was developed using the SpeechBrain toolkit (Ravanelli et al., 2024), and Hugging Face Transformers (Wolf et al., 2019).

Table 10: SLM results considering spoken content and acoustic elements using a subset of SALMon tasks.

| Tokenizer | #Q | Spoken Content | | | | Acoustic Consistency | | | Sem.-Ac. Align. |
|---|---|---|---|---|---|---|---|---|---|
| | | sBLIMP↑ | sWUGGY↑ | sSC↑ | tSC↑ | Gender↑ | Sent.↑ | Spk↑ | Sentiment↑ |
| HuBERT 25Hz | 1 | **60.89** | **70.51** | **53.23** | **71.46** | 69.50 | 62.50 | 69.00 | **53.00** |
| Enc-SMA-24 | 8 | 51.14 | 51.29 | 50.18 | 48.20 | 70.50 | 56.50 | 65.00 | 50.00 |
| DAC-SMA-16 | 8 | 51.51 | 50.73 | 48.95 | 51.52 | 81.00 | 60.00 | 77.00 | 50.00 |
| ST-S-16 | 8 | 51.08 | 56.89 | 48.42 | 55.74 | 66.50 | 58.00 | 65.00 | 49.50 |
| ST-S-16* | 8 | 52.75 | 63.46 | 47.56 | 60.60 | 67.00 | 59.50 | 65.50 | 50.00 |
| Mimi-S-24 | 8 | 52.25 | 62.21 | 51.52 | 54.30 | 77.50 | 71.50 | 78.00 | 52.00 |
| Mimi-S-24* | 8 | 60.17 | 67.57 | 51.68 | 68.51 | 76.50 | 77.00 | 76.00 | 52.00 |
| DWavL-S-16 | 6 | 53.96 | 69.10 | 51.41 | 62.42 | **92.00** | 70.00 | **86.50** | 49.00 |
| SQ-SMA-16 | 4 | 51.58 | 51.41 | 51.79 | 55.10 | 83.00 | 64.00 | 84.50 | 50.50 |
| WT-SMA-24 | 1 | 51.22 | 54.60 | 52.00 | 52.75 | 81.50 | **78.50** | 69.00 | 50.50 |

**Dataset.** We use the publicly available dataset LibriHeavy (Kang et al., 2024) containing 56k hours of transcribed speech, and the official validation and test sets of LibriSpeech (Panayotov et al., 2015).

**Evaluation Setup.** To evaluate our SLMs, we use the ZeroSpeech (Dunbar et al., 2021) sBLIMP and sWUGGY evaluation. The sBLIMP task assesses model perplexity on pairs of syntactically correct and incorrect sentences (e.g., *the dog sleeps* vs. *the dogs sleeps*). It evaluates the model's understanding of core grammatical phenomena in English. Similarly, sWUGGY measures whether the model assigns higher probability to a real word over a phonologically similar non-word (e.g., "brick" vs. "blick"), thus testing lexical discrimination. We further assess semantic understanding using Spoken Story-Cloze (sSC) and Topic Story-Cloze (tSC) tasks (Hassid et al., 2023), derived from the spoken variant of the StoryCloze dataset (Mostafazadeh et al., 2016). In sSC, the model must choose between the correct continuation and a randomly sampled, semantically incompatible adversarial one. This setup probes the model's ability to capture fine-grained causal and temporal commonsense relations. Similarly, in tSC, the adversarial continuation is taken from a different topic, so success reflects the model's capacity to maintain topical coherence.

To evaluate acoustic modeling such as prosody, speaker identity, and sentiment we adopt the SALMon evaluation suite (Maimon et al., 2025c). This includes several metrics that test whether the SLM retains and models key acoustic attributes. We focus on two aspects: (1) *acoustic consistency*, which evaluates the model's sensitivity to changes in speaker, gender, and sentiment; and (2) *sentiment-acoustic alignment*, which tests whether the model assigns higher scores to utterances where the acoustic sentiment aligns with the spoken content. This comprehensive suite allows us to assess both the linguistic and paralinguistic modeling capacities of our SLMs.

**Results and Discussions.** The results of the SLM experiments are presented in Table 10. To establish a clear baseline aligned with the current literature, we train an SLM using the HuBERT 25 Hz tokenizer, originally introduced in TWIST (Hassid et al., 2023), using the same configuration as previously described. Following the methodology of Défossez et al. (2024), all hybrid tokenizers marked with an asterisk ("*") correspond to SLMs trained with a semantic stream overweight factor of 100. This training setup prioritizes the semantic content over the acoustic content, and could be required for a better disentanglement of the semantic distillation. We maintain the same weighting strategy during evaluation.

The results reveal significant differences in performance across semantic and acoustic evaluation tasks. On semantically-oriented benchmarks (sBLIMP, sWUGGY, sSC, and tSC), most tokenizers exhibit limited semantic capacity. The tokenizer achieving the strongest semantic performance is HuBERT, followed by the semantically distilled weighted tokenizers that demonstrate the second-highest semantic performance. Specif-

ically, by overweighting the semantic stream, Mimi improves from 52.25% to 60.17% accuracy on sBLIMP. A similar trend is observed across all semantic evaluations, with an average performance gain of 6.91 accuracy points. The same pattern holds when comparing ST-S-16 to ST-S-16*, where the overweighted version achieves superior semantic results. Since semantic information is not clearly localized in a specific stream, WavLM is evaluated without any weighting strategy. Despite this, it still ranks second among unweighted tokenizers, just after HuBERT.

This semantic trend is consistent with our expectations of HuBERT, Mimi, SpeechTokenizer, and WavLM, since they primarily encode or have specific phonetic streams (via distillation or self-supervised learning objectives), thereby enhancing performance on linguistic understanding tasks. These results suggest that, when carefully tuned, semantic distillation approaches can rival SSL-based models like HuBERT, though they still fall slightly behind on sSC and tSC. Further investigation is needed to determine the optimal weighting between semantic and acoustic streams. In contrast, purely acoustic tokenizers such as Encodec and DAC show negligible semantic capability, rendering them unsuitable for this SLM configuration. Similarly, SQCodec and WavTokenizer offer limited semantic utility. We note that contrary to Encodec and DAC, SQCodec and WavTokenizer obtained stronger semantic scores. One explanation could be due to the type of quantization (i.e. RVQ vs. SVQ / FSQ) or the limited number of streams. Overall, the performance varies substantially across tokenizers, with only Mimi-S-24 approaching HuBERT's baseline on semantic tasks.

WavLM achieves the best acoustic performance on average, with accuracies of 92.00%, 70.00%, and 86.50% on gender, sentiment, and speaker consistency, respectively. This indicates that the model effectively captures and processes acoustic attributes in comparison to other tokenizers. Interestingly, Mimi also shows strong acoustic performance, outperforming HuBERT on each task. Purely acoustic tokenizers such as DAC and EnCodec, or SQ-codec and WavTokenizer also outperform HuBERT on acoustic evaluations and achieve results comparable to the semantically distilled tokenizers. However, no method yields substantial results on semantic-acoustic alignment, highlighting a limitation of current approaches in jointly modeling and reasoning over both modalities.

**Summary.** Our study reveals that semantic and acoustic performance in SLMs varies significantly across tokenizer types. HuBERT remains the strongest performer on semantic tasks, while WavLM leads in acoustic consistency. Semantically distilled tokenizers, particularly those with semantic stream overweighting, showed promising results by narrowing the semantic gap with HuBERT. These gains, however, come with trade-offs, emphasizing the importance of carefully balancing semantic and acoustic objectives. Overall, our findings suggest that, for now, there is no single tokenizer that excels across all spoken and acoustic tasks.

### 3.3.2 Text-to-Speech

**Background.** Text-to-Speech (TTS) is one of the primary applications of audio tokens. Traditional TTS systems typically rely on neural networks that predict mel spectrograms from text (Shen et al., 2018; Ren et al., 2019), followed by neural vocoders (Morise et al., 2016; Kong et al., 2020; van den Oord et al., 2016) to synthesize waveforms. The introduction of discrete audio tokens offers several advantages. Notably, it reframes waveform generation as a classification task over a fixed vocabulary, rather than a regression over continuous values. This shift enables optimization via categorical distributions and negative log-likelihood loss, which is typically more stable and tractable than regression objectives. Second, off-the-shelf neural codec decoders can reconstruct waveforms directly from token sequences, removing the need to train separate vocoders. Third, discrete tokens reduce sequence length, improving inference efficiency compared to µ-law quantization (van den Oord et al., 2016).

Prior to the adoption of discrete representations, TTS was largely dominated by non-autoregressive (NAR) models (Ren et al., 2021; 2019; Kim et al., 2021) due to their inference speed and stability relative to autoregressive (AR) models (Shen et al., 2018). However, the emergence of neural codecs has renewed interest in AR architectures (Wang et al., 2024c; Chen et al., 2025a; Yang et al., 2024c), which have demonstrated strong performance in expressive and zero-shot generation settings. Meanwhile, NAR models have also benefited from discrete token supervision, with recent advances incorporating diffusion- and flow-matching-based methods (Ju et al., 2024; Yang et al., 2024d) to further enhance synthesis quality.

**Experimental Setup.** Our models are based on a customized adaptation of the ESPnet (Tian et al., 2025) implementation of VALL-E (Chen et al., 2025a). To facilitate convergence, we adopt a staged training strategy that allows independent optimization of the AR and NAR components. We first perform 10 epochs of AR-only training, followed by 90 epochs of joint training with both AR and NAR layers to improve convergence for some tokenizers. All models use a 12-layer architecture for both AR and NAR decoders, with an attention dimension of 1024 and a dropout rate of 0.2. Following ESPnet conventions, we use looped nominal epochs, with 50,000 samples per epoch. The approach involves using a fixed number of data samples per epoch taken sequentially from the dataset until the end of the dataset is reached, at which point training restarts from the beginning. The epoch achieving the lowest validation dWER is chosen for evaluation.

**Dataset.** We train our model on the LibriTTS dataset (Zen et al., 2019), using corresponding phoneme transcriptions obtained from LibriTTS with Forced Alignment dataset (McAuliffe et al., 2017). The model is trained to generate discrete audio tokens conditioned on phoneme prompts (representing content) and acoustic code prompts (capturing target speaker characteristics), enabling it to synthesize speech that matches both the textual input and the target speaker's voice.

**Evaluation Setup.** We evaluate all models on all samples in the test split that fall within the length limit. To evaluate the speech naturalness of synthesized speech, we use a pretrained UTMOS model (Saeki et al., 2022). For assessing pronunciation accuracy, we transcribe both the ground-truth and generated utterances using the Whisper Large model (Radford et al., 2023) with greedy decoding. We compute the degraded Word-Error-Rate (dWER) by treating the ASR prediction of the ground-truth audio as the reference instead of the original transcription. We report mean UTMOS scores and micro dWER values; that is, the Levenshtein edit distance computed over the entire dataset. To measure speaker fidelity, we compute the cosine similarity (SpkSim) between X-vectors extracted from the generated and reference audio using the base variant of WavLM (Chen et al., 2022), fine-tuned for speaker verification.

Following ESPnet-Codec, to mitigate the variability of samples arising from using a text-conditioned sampling-based generative model, we generate 10 samples simultaneously for each prompt and choose the best one based on the WER calculated with the original label as the ground truth using Whisper Small, while final dWERs are calculated using the large model. To establish a clear baseline aligned with current literature, we train VALL-E using ESPNet's in-distribution retraining of EnCodec[11], which is trained on LibriTTS data only instead of a mix of speech, audio, and music as in the original model.

We perform a grid search over sampling temperature values $t \in 0.7, 0.8, 1.0, 1.2, 1.3$ and top-$k$ values $k \in 10, 20, 30$, where $t$ controls the sampling temperature and $k$ limits the number of highest-probability tokens considered during top-$k$ sampling. The optimal hyperparameters are selected based on the lowest dWER on a filtered subset of the validation set (67 samples selected from an initial random pool of 100, based on sequence length). These optimal values are then used for evaluation on the test set.

**Results and Discussion.** Table 11 presents the performance of various tokenizers on the TTS task. The highest audio quality is achieved by ESPNet EnCodec (Enc-S-24), which obtains a UTMOS score of 3.77. This model is trained on speech-only data, likely enabling it to better capture fine-grained speaker characteristics. Notably, the original EnCodec model performs significantly worse, with a UTMOS of only 2.31. We further investigate the impact of training data in Section 4.

The best adherence to the text is achieved with WavLM (Ji et al., 2024c) with a dWER of 4.32, which is likely attributable to the preservation of higher-level semantic information. It is followed by WavTokenizer at 4.67, which employs a single codebook with a larger vocabulary. The model also achieves a competitive audio quality at a UTMOS of 2.85. This result may be attributed to the expressive power of the larger vocabulary and the simplicity of single-stream generation.

Discrete WavLM6, achieves the second-best audio quality with a UTMOS of 3.42. It should be noted that during training, this tokenizer showed the most stable and robust results and early convergence, particularly in low-data regimes, and a reasonable speaker similarity at 0.90. These findings indicate that semantic rep-

---

[11]https://huggingface.co/espnet/libritts_encodec_24k

Table 11: Text-to-Speech synthesis results when using the VALL-E speech language model conditioned on phoneme annotations.

| Tokenizer | #Q | UTMOS↑ | dWER↓ | SpkSim↑ |
|-----------|-----|--------|-------|---------|
| Enc-SMA-24 | 8 | 2.31 | 4.77 | **0.91** |
| Enc-S-24 | 8 | **3.77** | 5.74 | **0.91** |
| DAC-SMA-24 | 8 | 2.47 | 11.71 | 0.88 |
| ST-S-16 | 8 | 2.91 | 5.35 | **0.91** |
| Mimi-S-24 | 8 | 2.60 | 7.93 | **0.91** |
| DWavL-S-16 | 6 | 3.42 | **4.32** | 0.90 |
| WT-SMA-24 | 1 | 2.85 | 4.67 | 0.88 |

resentations derived from self-supervised models are well-suited for TTS, supporting both natural-sounding and phonetically faithful speech synthesis.

In contrast, general-purpose acoustic tokenizers such as EnCodec (Enc-SMA-24) and DAC (DAC-SMA-24) result in decreased audio quality, and occasionally, as in the case of the latter, decreased pronunciation fidelity as well. These models likely require the TTS model to learn high-level speech abstractions from raw acoustic features, adding complexity to the generation process. Another possible contributing factor is that these tokenizers were trained on multi-domain audio data, whereas all other tokenizers evaluated were trained exclusively on speech. Finally, SQ-Codec, originally designed for diffusion-based generation with a large vocabulary ($\sim$ 20k), failed to converge in our AR/NAR setup, likely due to the challenges of modeling such a large token space in an autoregressive setting. All models achieve comparable levels of speaker similarity (ranging from 0.88 to 0.91).

**Summary.** Overall, achieving strong TTS performance with discrete tokenizers remains challenging, especially under constrained training conditions. Training with semantic tokenizers leads to more robust and effective TTS performance compared to acoustic or semantically distilled tokenizers; however, in high-data regimes with deep models, acoustic tokenizers, such as EnCodec, can be competitive with or even outperform semantic ones, particularly if they are trained on similar speech data, such as shown with Enc-S-24 trained on the same LibriTTS dataset as the TTS itself.

### 3.3.3 Audio Generation

**Background.** Generating realistic audio is a long-standing goal in generative AI, with applications in media, accessibility, and human-computer interaction. Previous works have explored audio generation under various conditions, including text (Liu et al., 2024b; Dong et al., 2023; Saito et al., 2024; Kumar et al., 2024), image (Sheffer & Adi, 2023; Wang et al., 2024a), video (Luo et al., 2023; Pascual et al., 2024; Zhang et al., 2024b; Wang et al., 2025b), and multimodal inputs (Jeong et al., 2025; Chen et al., 2025b), as well as in unconditional settings. In this study, we focus on both unconditional and text-conditioned generation, as these represent the most common and well-benchmarked paradigms.

Audio generation can be very broadly categorized into two main categories: diffusion-based methods (Yang et al., 2023b; Huang et al., 2023; Liu et al., 2023), and language model based approaches (Kreuk et al., 2023; Borsos et al., 2023b; Ziv et al., 2024). In this study, we focus on the latter, as the use of discrete audio tokens is more prevalent in such generation approaches. While both autoregressive (Kreuk et al., 2023; Borsos et al., 2023b) and non-autoregressive (Ziv et al., 2024) methods have been proposed for audio generation, we focus on AR models in this study. This choice is motivated by their typically superior generation quality and their prevalence in recent work, despite the trade-off of slower inference.

**Experimental Setup.** We adopt the AudioCraft toolkit (Copet et al., 2023), which provides a training pipeline for text-to-audio synthesis based on MusicGen (Copet et al., 2023) and AudioGen (Kreuk et al., 2023). The framework uses a T5 model (Raffel et al., 2020) to encode the text prompt into a latent conditioning tensor $\mathcal{C} \in \mathbb{R}^{T_{\text{len}} \times D}$. This tensor is passed to a causal decoder Transformer, where each block consists of a causal self-attention layer over previously generated audio tokens, followed by a cross-attention layer on the conditioning tensor. Following the original setup, we adopt a delay pattern across streams and apply Classifier-Free Guidance (CFG). We use the base model configuration with approximately 300M parameters and T5-Base as the text encoder. This framework is well-established and supports both multi-stream and single-stream audio tokens. To enable unconditional generation with the same model, we apply CFG dropout during 10% of training steps, a strategy shown to also enhance robustness in both conditional and unconditional settings.

We use the same architecture across all tokenizers to ensure a fair comparison. To balance consistency and computational efficiency, each model is trained for 100,000 steps using a batch size of 128 audio samples (each 10 seconds long), with mix-up augmentation. This corresponds to half the batch size and training steps used in the original AudioGen, while still achieving competitive performance. Note that the effective number of tokens and training time may vary depending on the tokenizer's frame rate and number of codebooks. For generation, we use fixed sampling parameters across all tokenizers, specifically top-k sampling with $k = 250$, without tuning them for individual models. Additional experiments confirm that the observed trends remain consistent under different sampling hyperparameters, though detailed results are omitted for brevity.

**Dataset.** We use several audio datasets for training and evaluation, many of which are part of *LAION-AUDIO-630K* (Wu et al., 2023c). Specifically, we include AudioCaps (Kim et al., 2019), AudioStock[12], BBC Sound Effects[13], EpidemicSound[14], FreeSound[15], Free to Use Sounds[16], MACS (Morato & Mesaros, 2021), and Odeon Sound Effects[17]. We follow Kreuk et al. (2023) and use the official splits of AudioCaps for validation and testing. All other datasets are used for training. These datasets vary in sample rate and format. We resample all audio and convert it to mono to match the input requirements of each tokenizer. In total, the training data contains approximately 4,050 hours of audio.

**Evaluation Setup.** We evaluate both text-conditioned generation and audio continuation with a 2.5 second audio prompt (and no text condition). We use three objective metrics that provide complementary perspectives for evaluation. First, we compute the Fréchet Audio Distance (FAD) using the FadTK toolkit (Gui et al., 2024) with the VGGish model. This metric is computed similarly to AudioGen (Kreuk et al., 2023), where FAD is calculated against the AudioCaps test set to measure the overall quality of synthesized audio. Second, we assess semantic consistency using KL Divergence. Specifically, we follow Yang et al. (2023b) and compare the output distribution of a pre-trained audio classifier, specifically PASST (Koutini et al., 2022), on real samples versus model-generated samples for the same conditions. Finally, we evaluate text-audio alignment using the CLAP score (Wu et al., 2023c; Huang et al., 2023). This measures how well the generated audio matches the input prompt. All evaluation metrics used here are generative in nature and are therefore influenced not only by the language model's ability to predict tokens but also by the quality of the vocoder used to synthesize audio. As such, poor metric scores may reflect limitations in the vocoder rather than issues in the encoder or language model.

**Results and Discussion.** Table 12 summarizes the performance for both text-conditioned generation and audio continuation. Since all Audio LM evaluations rely on generative outputs, final performance is often influenced by vocoder quality. As a result, even a language model with strong next-token prediction capabilities may underperform if paired with a suboptimal vocoder. To better isolate the effect of vocoding, we report reconstruction quality metrics for each tokenizer without involving any language model training. These results highlight notable differences across tokenizers. The 44kHz variant of DAC performs particularly

---

[12]https://audiostock.net/sfx
[13]https://sound-effects.bbcrewind.co.uk
[14]https://www.epidemicsound.com/sound-effects/
[15]https://freesound.org
[16]https://www.freetousesounds.com/all-in-one-bundle/
[17]https://www.paramountmotion.com/odeon-sound-effects

Table 12: Comparing performance of audio LMs over different tokenizers. We report FAD, KL divergence, and CLAP score on the AudioCaps test set. We also provide metrics for audio reconstruction. We note that ground truth audio in AudioCaps gets CLAP= 0.311, proving a topline. For more information about the tokenizers see Section 2.

| Tokenizer | #Q | Text Cond. Generation | | | Uncond. Generation | | | Reconstruction | | |
|---|---|---|---|---|---|---|---|---|---|---|
| | | FAD↓ | KLD↓ | CLAP↑ | FAD↓ | KLD↓ | CLAP↑ | FAD↓ | KLD↓ | CLAP↑ |
| Enc-SMA-24 | 8 | 3.771 | 1.555 | .279 | 5.996 | **1.897** | .202 | 3.806 | 0.456 | .281 |
| Enc-M-32 | 4 | 10.110 | 1.788 | .295 | 13.400 | 2.840 | .175 | 12.611 | 1.387 | .251 |
| Enc-A-16 | 4 | **1.955** | 1.576 | **.300** | **3.548** | 2.064 | .205 | **1.816** | 0.419 | .273 |
| DAC-SMA-44 | 9 | 6.929 | 1.959 | .267 | 6.732 | 2.041 | **.212** | 2.206 | **0.242** | **.299** |
| DAC-SMA-24 | 9 | 7.708 | 1.966 | .253 | 8.196 | 2.183 | .199 | 4.124 | 0.446 | .281 |
| SQ-SMA-16 | 4 | 7.733 | 3.078 | .151 | 5.977 | 2.301 | .175 | 3.460 | 0.460 | .268 |
| WT-SMA-24 | 1 | 2.594 | **1.463** | .291 | 4.441 | 2.224 | .193 | 5.018 | 0.892 | .253 |

well, reaching quality levels comparable to the version of EnCodec trained specifically on audio. In contrast, the music-only EnCodec model shows poor reconstruction quality, as expected given its domain-specific training. Apart from the music-only EnCodec, which was not trained on general audio, WavTokenizer exhibits the weakest reconstruction performance among all evaluated tokenizers.

The Audio LM trained on the music-only tokenizer (ENC-M-32) shows weak performance, particularly in terms of FAD. This may be attributed to limitations in the vocoder or the sensitivity of distribution-based metrics. For example, the model may be effective at next-token prediction and produce acoustically coherent samples, yet still deviate from the reference waveform distribution. The relatively strong CLAP and KLD scores for text-conditioned generation support this possibility, even though current evaluation metrics are insufficient to fully diagnose the cause of the performance drop. In contrast, the general-audio-trained version of EnCodec achieves the best FAD scores and consistently strong performance across all evaluation settings, emphasizing the value of domain-specific training for audio tokenizers.

WavTokenizer achieves strong performance in text-to-audio generation, despite its relatively poor reconstruction quality. One possible explanation is that its single-token stream format simplifies the modeling task for the language model, potentially enabling faster convergence. This performance gap may narrow with additional training compute. In contrast, both DAC variants and SQCodec exhibit weaker results in text-conditioned generation compared to their strong reconstruction performance. For SQCodec, the large and potentially redundant token vocabulary may make next-token prediction more difficult, reducing generation quality. Similarly, while the DAC models offer excellent compression, their structure may be more challenging for autoregressive modeling, resulting in reduced generation performance. This gap may be due to its higher token rate leading to longer sequences, or modeling difficulties specific to DAC.

**Summary.** Our findings highlight the critical role of domain-specific training for audio tokenizers. Training the language model alone on in-domain data is not sufficient: tokenizers must also be trained on the same domain to ensure strong performance. Our results also show that the best reconstruction performance does not correlate with the best modeling performance. In the future, we encourage the development of evaluation metrics that disentangle modeling ability from vocoder performance, as is common in the speech domain. We also emphasize the need for more robust modeling metrics (Chung et al., 2025).

### 3.3.4 Music Generation

**Background.** Music generation is particularly challenging due to the structural complexity of musical compositions, which involve diverse instrumentation, long-term dependencies, and high expectations for

both acoustic quality and aesthetic coherence. Recent advances in generative models, especially diffusion-based approaches and LLM-based architectures, have shown strong potential for producing coherent and high-quality music, often conditioned on melodic (Borsos et al., 2023b) or text (Huang et al., 2022) prompts.

A dominant paradigm in text-to-music generation involves latent diffusion models that operate over VAE-derived continuous representations (Evans et al., 2025; Chen et al., 2024a; Lam et al., 2023). In contrast, the use of autoregressive language models for music generation over discrete tokens remains an evolving area. Early work such as Jukebox (Dhariwal et al., 2020) employed VQ-VAE (van den Oord et al., 2017) to obtain quantized features for autoregressive modeling. More recent developments in neural audio codecs have shown that their latent spaces can serve as compact and expressive discrete representations of music. Building on this, several studies (Borsos et al., 2023b;a; Rouard et al., 2024) have explored LM-based music generation using various codec tokenizers and decoding strategies. In this section, we evaluate the effectiveness of discrete tokenizers in LM-based music generation, considering both text-conditioned synthesis and unconditional generation (i.e., music continuation).

**Experimental Setup.** Our experimental setup for text-to-music synthesis largely follows the same configuration described in the audio generation section. We use the AudioCraft toolkit (Copet et al., 2023) with a decoder-only transformer and cross-attention over a T5-Base (Raffel et al., 2020) text encoder. The model has approximately 300M parameters and is trained using classifier-free guidance (CFG). Key differences from the audio setup include the use of a higher CFG dropout rate of 30% (vs. 10% for audio), which allocates more emphasis to unconditioned (self-conditioned) music generation. Additionally, no mix-up augmentations are applied, and models are trained for 200,000 steps using $4\times$A100 40GB GPUs, each with a batch size of 32 samples (10 seconds each). This results in significantly less computing compared to the original MusicGen configuration (192 samples for 1M steps). As in the audio experiments, we use the same architecture across all tokenizers to ensure a fair comparison. We adopt autoregressive modeling, which, while less efficient than masked non-autoregressive methods (Garcia et al., 2023; Ziv et al., 2024), is more widely used and known to produce better perceptual quality.

**Datasets.** For training, we use the genre-balanced Free Music Archive (FMA) dataset (Defferrard et al., 2017), following the setup of stable-audio-open (Evans et al., 2025). All samples are 30 seconds long, and we follow the official split provided in the dataset repository[18]. The training set consists of 84,213 samples, totaling $\sim$702 hours. We combine artist, album, keywords, genres, and titles from the metadata to build the text prompt of the model.

For evaluation, we use two datasets: (1)MusicCaps (Agostinelli et al., 2023)[19], which contains 5,347 samples of 10 seconds each, annotated with descriptive text; and (2) the FMA test split, originally containing 11,235 samples of 30 seconds. To reduce evaluation time while preserving genre coverage, we randomly select 10 clips from each of the 156 genres, resulting in a genre-balanced subset of 1,560 samples.

**Evaluation Setup.** We evaluate music generation models on two tasks: text-conditioned generation and unconditional generation, also referred to as continuation, where a 2-second audio clip is used as a prompt to extend the content in a coherent manner. In addition, we assess the reconstruction performance of each tokenizer on both test datasets, providing an upper bound on the potential quality of generated outputs. Our evaluation protocol follows the same setup as in the text-to-audio experiments. Specifically, we use three objective metrics: FAD(Gui et al., 2024) computed with the VGGish model to assess audio quality, KLD between PASST classifier outputs(Koutini et al., 2022) for semantic consistency, and CLAP score (Wu et al., 2023c; Huang et al., 2023) to measure alignment with textual prompts.

**Results and Discussion.** The evaluation scores on MusicCaps and FMA test set are presented in Table 13. Surprisingly, the evaluation score on the MusicCaps is consistently better than those on the FMA-test, despite the theoretical similarity between the FMA-test and the FMA training data. Compared to MusicCaps, the FMA training set only provides very limited text prompts and compromised sound quality. However, thanks to its large dataset size and publicity, it is a natural choice for open-sourced model training

---

[18]https://github.com/mdeff/fma
[19]https://www.kaggle.com/datasets/googleai/musiccaps

Table 13: Comparing the performance of text-to-music LMs over different tokenizers on MusicCaps and FMA-test. The abbreviation of the tokenizer column are shown in Table 2. #Q denotes the number of quantization layers used in the tokenizer.

| Tokenizer | #Q | Text Cond. Generation | | | Uncond. Generation | | | Reconstruction | | |
|---|---|---|---|---|---|---|---|---|---|---|
| | | FAD↓ | KLD↓ | CLAP↑ | FAD↓ | KLD↓ | CLAP↑ | FAD↓ | KLD↓ | CLAP↑ |
| *MusicCaps* | | | | | | | | | | |
| Enc-SMA-24 | 8 | 11.173 | 2.246 | .108 | 4.632 | 0.904 | .275 | 2.209 | 0.259 | **.358** |
| Enc-M-32 | 4 | **4.264** | **2.006** | **.150** | **2.715** | 0.890 | **.282** | 1.995 | 0.356 | .339 |
| DAC-SMA-44 | 9 | 8.398 | 2.214 | .119 | 3.724 | **0.784** | **.282** | **0.927** | **0.182** | .340 |
| DAC-SMA-24 | 9 | 9.403 | 2.127 | .093 | 4.001 | 0.820 | .277 | 1.335 | 0.209 | **.358** |
| SQ-SMA-16 | 4 | 14.211 | 2.810 | .064 | 5.163 | 0.979 | .270 | 2.078 | 0.258 | .338 |
| WT-SMA-24 | 1 | 17.050 | 2.792 | .056 | 5.550 | 1.105 | .251 | 1.984 | 0.414 | .336 |
| *FMA* | | | | | | | | | | |
| Enc-SMA-24 | 8 | 15.380 | 2.161 | .059 | 14.478 | 1.827 | .065 | 1.013 | 0.287 | .141 |
| Enc-M-32 | 4 | 8.871 | **1.299** | **.078** | 8.357 | **1.006** | **.079** | 0.784 | 0.344 | .153 |
| DAC-SMA-44 | 9 | **8.115** | 1.543 | .062 | 6.398 | 1.100 | .075 | **0.494** | **0.196** | **.158** |
| DAC-SMA-24 | 9 | 8.789 | 1.746 | .039 | 7.002 | 1.405 | .043 | 0.708 | 0.222 | .125 |
| SQ-SMA-16 | 4 | 9.426 | 2.412 | .048 | **4.690** | 1.592 | .070 | 0.956 | 0.327 | .133 |
| WT-SMA-24 | 1 | 16.511 | 1.881 | .030 | 6.890 | 1.414 | .047 | 0.631 | 0.368 | .129 |

(Evans et al., 2025). We observe that models trained on FMA are to some extent adaptable to other datasets. We believe the quality of the FMA dataset explains the lower evaluation scores on the FMA-test. On this issue, we also want to call for efforts within this community to make prompt-rich text-to-music datasets publicly available.

Regarding reconstruction scores, EnCodec-32k tokenizer (despite being trained exclusively on music) does not consistently produce the highest quality. WavTokenizer achieves the best FAD score for reconstructed audio. DAC-44k and DAC-24k achieve the lowest KLD scores, indicating strong preservation of acoustic content. For text consistency, EnCodec-24k and DAC-24k perform best based on CLAP scores. Overall, DAC tokenizers at both sampling rates show the strongest reconstruction performance across metrics.

For text-conditioned generation, the music-specific tokenizer (EnCodec-32k) demonstrates clear advantages across all metrics and evaluation datasets. Among the multi-domain tokenizers, DAC-44k consistently outperforms DAC-24k, EnCodec-24k, and the single-stream WavTokenizer. This performance gap may stem from DAC-44k's higher bitrate, which likely enables it to produce a richer and more expressive representation, an essential factor for effective language modeling in music generation tasks.

For the unconditional generation task, EnCodec-32k and DAC-44k generally produce higher-quality outputs. An exception is observed on the FMA dataset, where SQ-Codec achieves better generation quality, as indicated by the FAD score. Across both datasets, unconditional generation consistently outperforms text-conditioned generation. We attribute this to the higher CFG dropout rate used during training, which exposes the models to more unconditioned scenarios and improves their ability to generate coherent continuations. Additionally, the poor quality of text prompts in the FMA dataset, also evident from its lower reconstruction scores, likely hinders the models' performance on text-conditioned tasks.

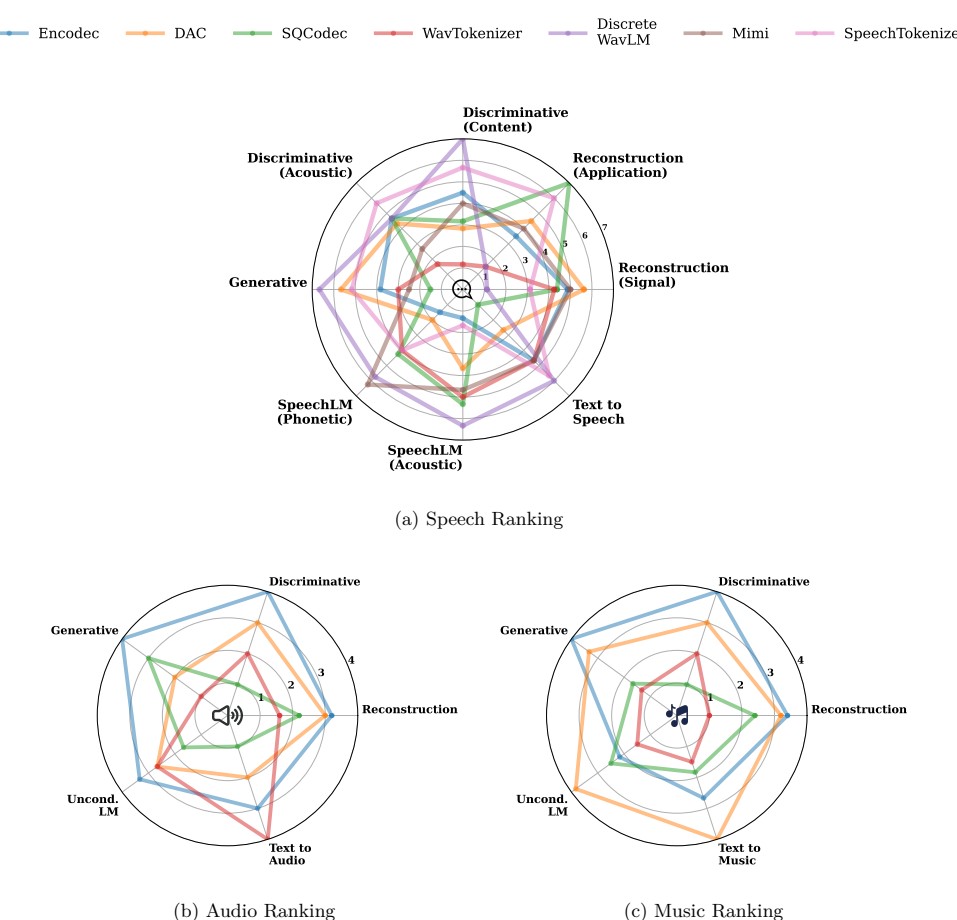

(a) Speech Ranking

(b) Audio Ranking

(c) Music Ranking

Figure 4: Average ranking of audio tokenizers across three domains: speech, general audio, and music.

**Summary.** For music LM, we observe that tokenizers with higher sample rates and multi-codebook, associated with higher bitrates, tend to perform better. This contrasts with audio and speech generation, where higher bitrate tokenizers were harder to model. We hypothesize that music, with its complex harmonic and temporal structure, benefits more from detailed representations, whereas such granularity may be excessive or less critical for general audio tasks. Additionally, unconditional generation consistently outperforms text-conditioned generation, emphasizing the benefits of providing melody prompts in music generation tasks.

### 3.4 General Trend

Figure 4 summarizes the overall ranking of audio tokenizers in three domains: speech, general audio, and music. These rankings are computed by first sorting the performance of each tokenizer per metric within each task (with rank 1 as worst and rank $N$ as best for $N$ tokenizers), then averaging the ranks across all tasks in the respective category. For tasks with multiple metrics, rankings are computed separately for each metric and then averaged. The resulting radar charts illustrate the average rank of each tokenizer across the following high-level categories, with higher values indicating better performance. For speech, we distinguish between two types of reconstruction: Reconstruction (Signal), which captures low-level fidelity (e.g., UTMOS), and Reconstruction (Application), which reflects the performance of resynthesized audio in downstream tasks (e.g., WER for ASR, speaker similarity for SV). Downstream tasks are categorized as either Discriminative or Generative. In speech, Discriminative tasks are further split into Content-level, which require higher-level semantic understanding (e.g., ASR, intent classification, keyword spotting), and Acoustic-level, which depend more on fine-grained acoustic cues (e.g., emotion recognition, speaker ID, speaker verification). Language

modeling tasks are grouped into two types: Text-conditioned and Unconditional (i.e., continuation-based generation). For speech, we include SpeechLM with separate subcategories for phonetic and acoustic metrics.

The radar charts highlight general trends in tokenizer strengths and weaknesses across domains. No tokenizer consistently outperforms others on all axes. The performance is strongly task- and domain-dependent. Some models excel at reconstruction but fall short in semantic modeling, while others achieve strong downstream results despite poorer signal fidelity. These plots are not meant to give strict recommendations, but rather to provide a high-level overview of performance trends. For real-world applications, we encourage referring to the full benchmark tables and task-specific analyses to identify the most appropriate tokenizer for the target use case.

## 4    Ablation Studies

While the above discussion has extensively evaluated publicly available discrete audio tokens across various applications, conducting fair comparisons between these codecs remains challenging. The development of audio tokenizers inherently involves numerous hyperparameters, including codebook setups, quantization algorithms, and training data composition, all of which significantly influence performance. This variability in model design and implementation creates substantial obstacles for researchers attempting to make meaningful comparisons, ultimately hindering both a comprehensive understanding of existing approaches and a systematic exploration of new audio tokenizer designs.

In this section, we aim to mitigate this issue by conducting experiments in a carefully controlled setup. Specifically, we use ESPnet-Codec (Shi et al., 2024c) as the base training framework to evaluate the effects of training data, codebook setups, quantization methods, and pre-trained model distillation.

### 4.1    Experimental Setups

To ensure reproducibility and isolate key variables in our experimental investigation, we establish a methodical framework that controls for potential confounding factors. First, we present our training data in various domains and sampling rates. Next, we establish uniform implementation protocols for model architecture and hyperparameter configuration, enabling direct performance comparisons across model variants. Building on this controlled foundation, we present a comprehensive set of experimental models, each systematically varying only the target parameters under investigation. Finally, we detail our evaluation methodology, including metrics and testing environments, to provide a consistent basis for assessing model performance and drawing meaningful conclusions about codec effectiveness.

**Training Dataset**. We prepare the dataset in three major domains, including speech, general audio, and music. All data in the three domains is sourced from the AMUSE dataset discussed in ESPnet-Codec (Shi et al., 2024c). The AMUSE dataset is a combination of high-quality datasets for codec training purposes. For speech, it contains DAPS, DNS Challenge 4, Commonvoice, VCTK, AISHELL3, Googlei18n-TTS corpora, and Mexican endangered languages (Dubey et al., 2024; Kuhn et al., 2014; Yamagishi et al., 2019; Shi et al., 2021a;b; Amith et al., 2021; Amith & López Francisco, 2022; Amith & Castillo Castillo, 2021; Ardila et al., 2020; Shi et al., 2021d). For general audio, it contains all data from the AudioSet unbalanced training set (Gemmeke et al., 2017). For music, it contains MusDB, Jamendo, OpenSinger, StyleSing111, M4Singer, Kiritan-singing, Oniku Kurumi Utagoe database, Natsume Singing database, Opencpop, ACE-KiSing (excluding original voices), PJS, and JSUT singing (Rafii et al., 2017; Bogdanov et al., 2019; Huang et al., 2021; Dai et al., 2023; Zhang et al., 2022; Ogawa & Morise, 2021; Wang et al., 2022; Shi et al., 2024a; Koguchi et al., 2020; Takamichi et al., 2020). To ensure equal consideration of the three domains and reduce the effect of unbalanced data distributions, we randomly sample 1k hours of data from each domain over the AMUSE dataset. The following experiments are conducted in either one of the domain-specific datasets or a combination of all three subsets. For each set of training data, we provide both 16 kHz version and 44.1 kHz version to consider the effect of different sampling rates.

**Model Implementation**. For our experiments, we utilize the Descript Audio Codec (DAC) framework implemented in ESPnet-Codec (Shi et al., 2024c; Kumar et al., 2023) as the foundation for neural-codec

Table 14: Summary of models used in the ablation study across 16kHz and 44.1kHz setups. Models are grouped by quantization method, RVQ, SVQ, FSQ, and Unit-based, with all models using the DAC backbone except D, which adopts Uni-HifiGAN. S, A, and M denote training on speech, general audio, and music, respectively. The "+" symbol indicates the use of distillation from SSL-based representations. Checkmarks (✓) indicate domain-specific training data used for each model variant.

| Model | Base | Quantization | Distillation | Data Domains | | |
|-------|------|--------------|--------------|--------|-------|-------|
| | | | | Speech | Audio | Music |
| RVQ-S | | | - | ✓ | | |
| RVQ-S+ | | | ✓ | ✓ | | |
| RVQ-A | DAC | RVQ | - | | ✓ | |
| RVQ-M | | | - | | | ✓ |
| RVQ-3 | | | - | ✓ | ✓ | ✓ |
| SVQ-S | | | - | ✓ | | |
| SVQ-S+ | | | ✓ | ✓ | | |
| SVQ-A | DAC | SVQ | - | | ✓ | |
| SVQ-M | | | - | | | ✓ |
| SVQ-3 | | | - | ✓ | ✓ | ✓ |
| FSQ-S | | | - | ✓ | | |
| FSQ-A | DAC | FSQ | - | | ✓ | |
| FSQ-M | | | - | | | ✓ |
| FSQ-3 | | | - | ✓ | ✓ | ✓ |
| K-means-S | Unit-HifiGAN | K-means | - | ✓ | | |

training. To evaluate discrete SSL approaches, we employ the discrete unit-based HiFiGAN vocoder as implemented by Yan et al. (2023).

Our ablation studies focus on two key aspects: quantization techniques and semantic distillation. For quantization, we compare three distinct approaches: RVQ, single-layer VQ, and FSQ. Additionally, following recent research demonstrating the efficacy of self-supervised learning representations as distillation targets for quantizer training (see Section 2 for more discussion), we incorporate semantic distillation variants for both our RVQ and VQ-based models to systematically evaluate its impact.

**Summary of Models**. Candidate models are summarized in Table 14. For each model listed in the table, we conduct the training at 16 kHz and 44.1 kHz with the corresponding dataset. Here, we mostly follow the previous literature discussed in Section 2, where we ignore model setups with no or limited related work, such as the scenarios of using SSL-based distillation in audio or music domains or the use of Uni-HifiGAN for higher sampling rates. While we standardize core training parameters across all experimental conditions to ensure fair comparisons, we implement targeted customization for specific model variants to integrate different ablation factors. Complete documentation of both the standardized parameters and model-specific adjustments is available in our released model checkpoints[20].

**Evaluation Setup**. We follow the reconstruction evaluation protocol outlined in Section 3.1, adapting our methodology to accommodate the specific requirements of different data domains. For evaluation data, we utilize the LibriSpeech test-clean set (speech), the AudioSet test set (general audio), and the MUSDB test set (music).

---

[20]https://huggingface.co/collections/espnet/codec-survey-pre-trained-models-67ce8e09568b741d1c4483c8

Table 15: Ablation experiments on audio tokenizer reconstruction performance (speech domain).

| Model\SR(kHz) | SDR↑ | | SI-SNR↑ | | PESQ↑ | | UTMOS↑ | | DNSMOS P835↑ | | WER↓ | | Spk Sim↑ | |
|---|---|---|---|---|---|---|---|---|---|---|---|---|---|---|
| | 16 | 44.1 | 16 | 44.1 | 16 | 44.1 | 16 | 44.1 | 16 | 44.1 | 16 | 44.1 | 16 | 44.1 |
| Ground truth | - | - | - | - | - | - | 4.09 | 4.09 | 3.18 | 3.18 | 2.83 | 2.83 | - | - |
| RVQ-S | 4.08 | 8.24 | 1.15 | 6.38 | 2.59 | 3.24 | 3.35 | 3.62 | 3.16 | 3.16 | 2.04 | 2.63 | 0.67 | 0.90 |
| RVQ-S+ | 1.63 | 8.40 | -1.77 | 5.67 | 2.22 | 3.30 | 3.12 | 3.65 | 3.12 | 3.14 | 2.12 | 3.17 | 0.69 | 0.89 |
| RVQ-A | 0.59 | 6.92 | -3.36 | 4.27 | 2.02 | 2.86 | 1.81 | 3.15 | 2.58 | 3.06 | 2.47 | 2.92 | 0.51 | 0.73 |
| RVQ-M | 2.80 | 6.74 | -0.68 | 4.61 | 2.00 | 2.41 | 1.64 | 2.33 | 2.64 | 2.68 | 3.20 | 2.85 | 0.44 | 0.56 |
| RVQ-3 | 2.46 | 7.50 | -1.12 | 4.63 | 2.43 | 3.06 | 2.71 | 3.33 | 2.96 | 3.08 | 2.22 | 2.66 | 0.61 | 0.87 |
| SVQ-S | -4.90 | 0.92 | -13.59 | -2.04 | 1.43 | 1.69 | 2.19 | 2.61 | 3.09 | 3.10 | 13.16 | 8.28 | 0.35 | 0.53 |
| SVQ-S+ | -4.45 | -0.64 | -11.74 | -3.64 | 1.42 | 1.63 | 2.14 | 2.40 | 3.00 | 3.07 | 13.71 | 7.89 | 0.36 | 0.52 |
| SVQ-A | -11.80 | -5.04 | -33.46 | -9.56 | 1.19 | 1.18 | 1.25 | 1.26 | 1.86 | 1.97 | 31.40 | 34.68 | 0.19 | 0.14 |
| SVQ-M | -5.19 | -4.70 | -11.17 | -7.89 | 1.20 | 1.14 | 1.29 | 1.24 | 2.19 | 1.56 | 14.88 | 33.87 | 0.15 | 0.11 |
| SVQ-3 | -5.90 | -3.09 | -15.13 | -7.12 | 1.29 | 1.79 | 1.44 | 2.57 | 3.10 | 3.01 | 22.24 | 6.71 | 0.26 | 0.49 |
| FSQ-S | 3.89 | 4.58 | 1.75 | 2.47 | 2.08 | 2.10 | 3.29 | 3.06 | 3.21 | 3.12 | 3.57 | 4.02 | 0.48 | 0.66 |
| FSQ-A | 1.14 | 1.00 | -1.89 | -1.58 | 1.94 | 1.74 | 2.84 | 2.41 | 3.15 | 2.97 | 5.12 | 7.59 | 0.43 | 0.39 |
| FSQ-M | -1.08 | -2.09 | -4.39 | -4.61 | 1.39 | 1.22 | 1.57 | 1.26 | 2.66 | 2.04 | 24.46 | 20.26 | 0.17 | 0.16 |
| FSQ-3 | 2.41 | 1.29 | 0.01 | -1.28 | 1.97 | 1.79 | 3.06 | 2.57 | 3.20 | 3.01 | 4.35 | 6.71 | 0.44 | 0.49 |
| K-means-S | -18.21 | - | -42.98 | - | 1.05 | - | 2.28 | - | 2.46 | - | 6.78 | - | 0.13 | - |

Table 16: Ablation experiments on audio tokenizer reconstruction performance (audio and music domain).

| | Audio | | | | | | | | | | Music | | | | | | | | | |
|---|---|---|---|---|---|---|---|---|---|---|---|---|---|---|---|---|---|---|---|---|
| | SDR↑ | | CI-SDR↑ | | SI-SNR↑ | | VISQOL↑ | | SingMOS↑ | | SDR↑ | | CI-SDR↑ | | SI-SNR↑ | | VISQOL↑ | | SingMOS↑ | |
| Model\SR(kHz) | 16 | 44.1 | 16 | 44.1 | 16 | 44.1 | 16 | 44.1 | 16 | 44.1 | 16 | 44.1 | 16 | 44.1 | 16 | 44.1 | 16 | 44.1 | 16 | 44.1 |
| RVQ-S | -4.05 | 2.85 | -4.02 | 2.51 | -10.07 | -0.02 | 4.18 | 3.81 | 2.59 | 2.66 | 0.45 | 6.80 | 0.44 | 6.75 | -2.29 | 4.83 | 4.21 | 4.17 | 2.67 | 2.60 |
| RVQ-S+ | -8.90 | 2.52 | -8.83 | 2.19 | -17.57 | -1.14 | 4.13 | 3.79 | 2.57 | 2.63 | -6.73 | 6.60 | -6.70 | 6.55 | -11.10 | 4.46 | 4.07 | 4.24 | 2.64 | 2.70 |
| RVQ-A | -0.59 | 3.65 | -0.61 | 3.21 | -5.56 | 0.48 | 4.22 | 3.78 | 2.63 | 2.65 | 5.78 | 8.24 | 5.70 | 8.17 | 2.87 | 5.97 | 4.21 | 4.12 | 2.71 | 2.72 |
| RVQ-M | 0.65 | 3.76 | 0.60 | 3.35 | -4.07 | 1.06 | 4.13 | 3.62 | 2.63 | 2.63 | 6.34 | 8.33 | 6.25 | 8.26 | 3.82 | 6.48 | 4.13 | 3.99 | 2.70 | 2.68 |
| RVQ-3 | 0.14 | 3.99 | 0.11 | 3.50 | -4.47 | 0.70 | 4.17 | 3.83 | 2.61 | 2.65 | 5.75 | 8.54 | 5.68 | 8.46 | 3.32 | 6.34 | 4.12 | 4.07 | 2.68 | 2.72 |
| SVQ-S | -14.80 | -8.43 | -14.72 | -8.07 | -31.86 | -14.73 | 3.83 | 3.28 | 2.55 | 2.59 | -14.25 | -4.78 | -14.22 | -4.76 | -27.16 | -7.77 | 3.68 | 3.64 | 2.59 | 2.70 |
| SVQ-S+ | -14.74 | -9.39 | -14.66 | -9.00 | -30.69 | -16.37 | 3.84 | 3.23 | 2.54 | 2.59 | -15.10 | -5.24 | -15.06 | -5.23 | -27.69 | -8.80 | 3.69 | 3.60 | 2.58 | 2.68 |
| SVQ-A | -13.08 | -6.80 | -13.00 | -6.46 | -30.70 | -12.80 | 3.86 | 3.32 | 2.56 | 2.59 | -6.54 | -0.49 | -6.52 | -0.40 | -14.76 | -3.21 | 3.68 | 3.52 | 2.65 | 2.69 |
| SVQ-M | -6.53 | -5.50 | -6.47 | -5.19 | -13.43 | -10.62 | 3.94 | 3.33 | 2.59 | 2.58 | -0.35 | 0.52 | -0.35 | 0.52 | -2.94 | -1.80 | 3.87 | 3.33 | 2.70 | 2.63 |
| SVQ-3 | -9.28 | -7.47 | -9.21 | -7.12 | -19.50 | -14.32 | 3.88 | 3.34 | 2.52 | 2.62 | -2.75 | -1.35 | -2.73 | -1.35 | -6.75 | -4.63 | 3.72 | 3.54 | 2.63 | 2.73 |
| FSQ-S | -7.32 | -4.26 | -7.26 | -4.04 | -14.22 | -8.12 | 4.05 | 3.32 | 2.60 | 2.63 | -5.04 | -0.37 | -5.02 | -0.37 | -8.42 | -2.84 | 3.80 | 3.65 | 2.69 | 2.71 |
| FSQ-A | -2.79 | -2.00 | -2.76 | -1.00 | -7.05 | -5.37 | 4.09 | 3.53 | 2.63 | 2.65 | 0.90 | 2.62 | 0.80 | 2.60 | -1.14 | 0.49 | 4.00 | 3.92 | 2.70 | 2.73 |
| FSQ-M | -3.37 | -3.25 | -3.33 | -3.05 | -8.23 | -6.85 | 4.02 | 3.41 | 2.62 | 2.60 | 1.54 | 2.70 | 1.52 | 2.77 | -0.67 | 0.66 | 3.99 | 3.54 | 2.71 | 2.66 |
| FSQ-3 | -3.22 | -2.36 | -3.19 | -2.24 | -7.86 | -6.01 | 4.06 | 3.53 | 2.61 | 2.62 | 0.89 | 2.75 | 0.88 | 2.73 | -1.38 | 0.19 | 3.96 | 3.74 | 2.69 | 2.68 |
| K-means-S | -21.01 | - | -20.89 | - | -47.37 | - | 3.14 | - | 2.78 | - | -19.53 | - | -19.49 | - | -46.03 | - | 2.82 | - | 2.87 | - |

Sampling rate considerations necessitated domain-specific evaluation approaches. For speech reconstruction, all evaluations are conducted at 16 kHz, even when testing 44.1 kHz neural codecs, to maintain consistency with the source LibriSpeech dataset. For music evaluation, we use different sampling rates based on model capabilities: 16 kHz for models designed at that native rate, and 24 kHz for 44.1 kHz neural codecs, reflecting the upper-frequency limitations in the MUSDB test set. For general audio evaluation, we align the evaluation with the codec models. These adjustments ensure fair comparisons while respecting both technical constraints and the inherent characteristics of each dataset.

## 4.2 Results and Discussion

The results of our ablation study are shown in Table 15 and Table 16. We summarize our findings as follows:

**Data Domains**. Domain alignment between training and testing data has emerged as a critical determinant of performance in discrete audio representation modeling. Our experiments confirm that reconstruction quality consistently peaks when models are evaluated on domains matching their training data. More significantly, we observe that even with carefully balanced multi-domain training datasets, models still exhibit notable performance degradation when assessed on individual domains compared to domain-specific training. These challenges have become increasingly relevant in light of recent audio foundation models, which aim for broad generalization across diverse audio types. Our findings highlight the need for two crucial research directions: *developing more effective methodologies for balancing domain-specific optimization*, and *addressing the fundamental challenges of cross-domain generalization in discrete audio representation learning.*

**Sampling Rate**. While prior research has rarely examined sampling rate effects on discrete audio representation, this gap is largely due to methodological challenges in creating controlled comparisons across different rate conditions. Our systematic ablation study addresses this limitation and reveals sampling rate as a significant factor in model performance. As shown in Table 15 and Table 16, RVQ-based models consistently demonstrate performance improvements across multiple evaluation metrics when trained at 44.1 kHz, even when the reconstructed audio is downsampled to 16 kHz for evaluation. These benefits, however, are not universal across all quantization approaches. Models utilizing Finite Scalar Quantization (FSQ) actually exhibited performance degradation on several metrics when trained at higher sampling rates. This contrasting behavior indicates that the relationship between sampling rate and model effectiveness is contingent on the specific quantization methodology employed. Based on these findings, we recommend that future research on discrete audio representation should incorporate sampling rate as a critical design parameter, with careful optimization based on the selected quantization approach and target application domain.

**Distillation Effect**. Prior studies involving distillation from pre-trained models have frequently demonstrated that such distillation supports comparable or improved signal reconstruction, while providing substantial performance benefits for downstream tasks (Du et al., 2023; Défossez et al., 2024; Zhang et al., 2024a). In our controlled ablation analyses, we similarly observed that incorporating distillation from pre-trained speech representations can enhance model performance on certain metrics for signal reconstruction, as demonstrated by our comparison between models trained with and without distillation (e.g., A-S vs. A-S+). However, it should be noted that the domain-specific nature of the pre-trained model may limit generalization, especially when applied to broader domains such as general audio or music, as evidenced in Table 16. This highlights a potential trade-off between achieving high performance in specialized tasks and maintaining broader generalization capabilities.

**Quantization Methods**. Our experiments demonstrate that different quantization methods significantly impact codec performance. The RVQ modeling consistently outperforms other quantization approaches across most reconstruction metrics. Conversely, SVQ models typically yield the poorest results. An interesting exception emerges with FSQ at 16 kHz, which surpasses RVQ in speech quality metrics measured by UTMOS and DNSMOS. Generally, RVQ demonstrates a higher potential for audio quality due to its high-fidelity reconstruction capabilities. However, we caution readers that the performance alignment between audio reconstruction and downstream applications is not guaranteed. The metrics observed in this study may not directly translate to broader application performance, and further research is needed to establish definitive correlations.

## 5 Conclusion and Future Directions

Discrete units offer several advantages in audio representation. They provide compact, modular, and scalable abstractions that are particularly well-suited for generative tasks such as speech synthesis, music generation, and general audio modeling. By converting regression-based waveform modeling into classification tasks, discrete tokenization simplifies both training and inference. These representations are also more efficient in terms of storage and transmission, making them advantageous for resource-constrained deployment and streaming applications. Additionally, discrete tokens align naturally with large language model architectures, facilitating integration into multimodal systems. While discrete representations may currently underperform continuous features on certain discriminative tasks, particularly in low-resource or semantically fine-grained scenarios, recent advances show that they can match or even outperform continuous representations in some

applications, such as text-to-speech, especially when trained on large-scale data. These trends highlight the growing promise of discrete audio tokenization and motivate continued research to improve their robustness, expressiveness, and generalization across diverse downstream tasks.

Based on our comprehensive analysis, we identify several key observations and open challenges in the design, evaluation, and application of discrete audio tokenizers. These point to promising future research directions:

- **Scaling Limitations and Generalizability**: While our experiments used moderately sized models and datasets, this choice was intentional to ensure fair and controlled comparisons across tokenizers. These settings reflect realistic constraints for many academic and open-source efforts. Importantly, our key findings, such as the superior performance of semantic over acoustic tokenizers for semantic tasks, are aligned with trends observed in larger-scale systems, suggesting that these insights are likely to hold as models and datasets scale up.

- **Correlation between Reconstruction and Downstream Performance**: We observed a clear trade-off between high-fidelity signal reconstruction and downstream task performance. Optimizing tokenizers purely for reconstruction often fails to preserve task-relevant features such as phonetics or semantic content. This is especially evident in tasks where the decoder is not involved (e.g., ASR or SLU). Future research should aim to jointly optimize for both signal fidelity and semantic utility, possibly using multi-task or adversarial training.

- **Fair and Consistent Evaluation**: Tokenizers vary significantly in training data, sampling rate, and domain scope (e.g., speech-only vs. multi-domain). These discrepancies complicate benchmarking. Our study highlights the importance of standardizing evaluation pipelines to allow fair comparison across tokenizers. Establishing unified benchmarks with consistent experimental settings remains an urgent need.

- **Benchmark vs. Reported Performance Gap**: Results reproduced under controlled benchmark settings often fall short of the originally reported numbers. This indicates that some improvements reported in prior works may rely on favorable hyperparameter tuning or large-scale resources. There is a need for reproducibility-focused evaluations and scaling studies to better understand real-world tokenizer performance.

- **Semantic Distillation Beyond Speech**: Most existing distillation-based tokenizers focus exclusively on speech. Extending semantic distillation techniques to music and general audio domains is an underexplored direction that could substantially improve discrete token quality across diverse audio tasks.

- **Discrete vs. Continuous Representations**: While discrete tokenizers have shown promising progress, continuous representations often remain superior for speech-language understanding tasks that rely on preserving fine-grained acoustic cues such as prosody, emotion, and speaker traits. However, this performance gap may not be universal. For instance, discrete tokens can be more suitable in settings involving autoregressive or masked generative modeling, where classification-based objectives are advantageous. Conversely, models based on score matching or diffusion may still benefit from continuous conditioning inputs. Bridging these differences remains an open challenge for integrating audio tokenizers into multimodal LLMs that require semantic richness.

- **Toward Unified Tokenizers**: Future systems may benefit from unified tokenizers that can support both generative and discriminative tasks across multiple audio domains. Achieving this will likely require architectures that balance streamability, semantic alignment, reconstruction quality, and domain generalization, possibly via modular or hierarchical designs.

- **Trustworthiness**: Trustworthy concerns such as bias (Ren et al., 2024a) and deepfakes (Wu et al., 2024e; Du et al., 2025) are required to be considered. Modern discrete-unit-based speech generation can mimic voices with human-like realism (Chen et al., 2025a), raising risks of misuse by malicious actors—e.g., generating fake news using a public figure's voice.

In summary, while some questions about audio tokens remain open, our evaluation provides a comprehensive perspective that highlights general trends, key challenges, and guidelines for selecting a tokenizer. We hope our study offers valuable insights to the research community and helps pave the way for future advancements in this rapidly progressing field.

**Author Contributions**

All core members participated equally in the project. While this section outlines the primary responsibilities of each core contributor, many also contributed to other aspects of the project. Pooneh Mousavi led the overall survey and benchmarking effort. Haibin Wu, Pooneh Mousavi, and Anastasia Kuznetsova co-led the design of the tokenizer taxonomy, with collaboration with Haici Yang and Gallil Maimon. Haibin Wu, Jiatong Shi, and Darius Petermann conducted the reconstruction experiments. Pooneh Mousavi led the development of the downstream section, with contributions from Darius Petermann, and Anastasia Kuznetsova. Adel Moumen led the development of the SpeechLM evaluations. Artem Ploujnikov was responsible for the TTS evaluation experiments. Gallil Maimon conducted the AudioLM experiments. Haici Yang led the MusicLM evaluation. Jiatong Shi performed the ablation study in controlled setup. All authors participated in writing, editing, and refining the paper. The advisors provided guidance and critical feedback throughout the project.

**Acknowledgments**

We thank Jinchuan Tian for valuable discussions with TTS. We also greatly thank Dongchao Yang for sharing his in-depth, hands-on experience with SQ-Codec development and thoughts about the tokenizer taxonomy design, which helped clarify important aspects of our survey. We are grateful to Huizhong Lu for computing support. We thank the NVIDIA Academic Grant Program for donating GPU hours used for this project. This work was supported by the Cambridge Commonwealth, European & International Trust (scholarship to Adel Moumen); the Natural Sciences and Engineering Research Council of Canada (NSERC); the Digital Research Alliance of Canada (alliancecan.ca); The Israel Science Foundaton, grant 2049/22 (scholarship to Gallil Maimon); an Amazon Research Award (ARA); Electronics and Telecommunications Research Institute (ETRI) grant funded by the Korean government [24ZC1100, The research of the basic media/contents technologies]. We also thank Jean Zay GENCI-IDRIS for their support in computing (Grant 2024-AD011015344 and Grant 2024–A0161015099). Cem Subakan is supported by Discovery Grant RGPIN 2023-05759.

Part of the experiments used the Bridges2 at PSC and Delta/DeltaAI NCSA computing systems through allocation CIS210014 from the Advanced Cyberinfrastructure Coordination Ecosystem: Services & Support (ACCESS) program, supported by National Science Foundation grants 2138259, 2138286, 2138307, 2137603, and 2138296.

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

# A  Additional Evaluation for Reconstructed Audio Quality

Table 17 shows the UTMOS V2 scores for different audio tokenizers.

Table 17: UTMOS V2 scores of audio tokenizers across different bitrates.

| Model | Bitrate (kbps) | UTMOS V2 |
|---|---|---|
| Enc-SMA-24 | 1.5
6.0
24.0 | 1.40
2.13
2.71 |
| DAC-SMA-24 | 1.5
6.0
24.0 | 1.40
2.46
3.06 |
| ST-S-16 | 1.0
4.0 | 1.52
2.92 |
| Mimi-S-24 | 1.1
4.4 | 2.12
2.82 |
| DWavL-S-16 | 1.0
3.0 | 2.16
2.33 |
| SQ-SMA-16 | 3.0 | 3.19 |
| WT-SMA-24 | 0.98 | 2.39 |
| WT-S-24 | 0.52 | 3.12 |

# B  Computational Setup

Table 18 shows computaional setting for our experiments. Approximate runtimes are provided per run and may vary slightly between tokenizers depending on token rate, number of token streams, and other factors. For downstream tasks time is reported per run, runtimes range from 2 hours (e.g., Keyword Spotting) to 48 hours (e.g., ASR), depending on the task complexity.

Table 18: Computational settings for our experiments.

| Experimental Name | Computational Resource | Approx. Time |
|---|---|---|
| Downstream Evaluation | 1×A100 (80GB) | 2–48 hrs |
| Reconstructed Audio Quality | 1×A6000 (48G) | 24 hrs |
| Speech Language Modeling | 2×A100 (40GB) | 48 hrs |
| Text-to-Speech | 1×A100 (80GB) | 96 hrs |
| Audio Generation | 2×A100 (80GB) | 48 hrs |
| Music Generation | 4×A100 (40GB) | 48 hrs |
| Ablation Studies | 2×GH200 (100GB) | 48 hrs |

# C  Dataset

For each experiment, we report the dataset, source, and approximate duration of the corresponding dataset. A detailed summary is provided in Table 19. For more information, please refer to the "Dataset" subsection in each benchmark section.

Table 19: Summary of datasets, their specifications, and approximate number of hours used for all experiments in this survey.

| Task | Dataset | ~Hours | Data Link |
|---|---|---|---|
| *Reconstruction* | | | |
| Speech | LibriSpeech (Korvas et al., 2014) | 6 | Link |
| Music | MUSDB (Rafii et al., 2017) | 10 | Link |
| Audio | Audioset (Gemmeke et al., 2017) | 56 | Link |
| *Downstream* | | | |
| ASR (En) | LibriSpeech (Korvas et al., 2014) | 1,000 | Link |
| ASR (Welsh) | CommonVoice 17.0 (Ardila et al., 2020) | 8 | Link |
| ASR (Basque) | CommonVoice 17.0 (Ardila et al., 2020) | 116 | Link |
| Speaker ID / Verification | VoxCeleb1 (Nagrani et al., 2017) | 350 | Link |
| Emotion Recognition | IEMOCAP (Busso et al., 2008) | 7 | Link |
| Keyword Spotting | Speech Commands (Warden, 2018) | 18 | Link |
| Intent Classification | SLURP (Bastianelli et al., 2020) | 10 | Link |
| Speech Enhancement | VoiceBank (Valentini-Botinhao et al., 2016) | 10 | Link |
| Speech Separation | Libri2Mix (Cosentino et al., 2020) | 400 | Link |
| Music Genre Classification | GTZAN (Tzanetakis & Cook, 2002) | 8 | Link |
| Music Source Separation | MUSDB (Rafii et al., 2017) | 10 | Link |
| Sound Event Classification | ESC-50 (Piczak, 2015) | 2 | Link |
| Audio Separation | FUSS (Wisdom et al., 2021) | 23 | Link |
| *Acoustic LM* | | | |
| Speech Language Modeling | LibriHeavy (Kang et al., 2024) | 56,000 | Link |
| Text-to-Speech | LibriTTS (Zen et al. (2019)) | 960 | Link |
| Audio Generation | Data Mix (see Sec. 3.3.3 for details) | 4050 | - |
| Music Generation | FMA (Defferrard et al., 2017) | 702 | Link |
| *Ablation* | | | |
| Speech | Data Mix (see Sec. 4 for details) | 1,000 | - |
| Music | Data Mix (see Sec. 4 for details) | 1,000 | - |
| Audio | Data Mix (see Sec. 4 for details) | 1,000 | - |

