# OpenReview forum: "Discrete Audio Tokens: More Than a Survey!"
_TMLR — Accepted by TMLR_

### Review · Reviewer_ZrAE · 2025-07-04

**Summary Of Contributions:**

The contributions by this paper are threefold.

1. This paper presented a systematic review and benchmark of audio tokenizers for speech, music, and general audio signals.

2. This paper proposed a taxonomy of tokenization approaches based on the DNN architectures, quantization techniques, training paradigm, streamability, and application domains.

3. This paper evaluated audio tokenizers on multiple benchmarks for reconstruction, downstream performance, and audio language modeling. It also analyzed trade-offs through ablation studies.

In summary, the findings by this paper highlight key limitations, practical considerations, and open challenges that provide insight and guidance for future research in audio tokenizers and their applications in audio language modeling. I really appreciate the authors' effort to conduct such large-scale experimental investigations. I believe that this paper should be valuable for research community in not only speech/audio but also machine learning research, with some revisions to improve the presentation and reproducibility.

**Audience:**

Yes

**Claims And Evidence:**

No

**Requested Changes:**

[Major comments]

[C1] As this is the journal named as Transactions on "Machine Learning Research," it would be better to include discussion of compared codec models and evaluation results from the viewpoints of machine learning. For example:
- [C1-1] The codebook collapse is well-known issue in VQVAE-based generative models. Does this issue also occur in the context of neural audio codecs and affect the quality of generated samples?
- [C1-2] Please add explanation with mathematical formulations in some parts, such as "Masked Prediction (MP)" in Section 2.4.2 and "Disentanglement" in Section 2.4.3.
- [C1-3] In Section 2.4.2 "Adversarial (GAN)," please explain what the "perceptual losses" are, with proper mathematical formulation.

[C2] The following terms should be explained with proper definition, as not all readers are familiar with these technologies:
- "acoustic language model(s)"
- "self-supervised learning (SSL) model(s)" for speech/audio data

[C3] Please clarify the following points regarding the experimental setups and evaluation results:
- [C3-1] Why did the authors choose the tokenizers listed in Tables 2 and 14 for their experiments from various ones listed in Table 1?
- [C3-2] Why did the authors employ multiple models to estimate the perceptual quality of generated samples?
  - As the UTMOS was developed in 2022, it might not be suitable for evaluating the quality of synthetic speech produced by recent neural speech synthesis models like VALL-E. How about using a newer MOS prediction model, such as UTMOSv2[a]?
- [C3-3] For all experiments please describe the computational resources, i.e., the number of GPUs and their device names.

[a] K. Baba et al., "The T05 System for The VoiceMOS Challenge 2024: Transfer Learning from Deep Image Classifier to Naturalness MOS Prediction of High-Quality Synthetic Speech," in Proc. SLT 2024.

[C4] The following claims should be explained in more detail:
- [C4-1] In Section 3.4: "For real-world applications, we encourage referring to the full benchmark tables and task-specific analyses to identify the most appropriate tokenizer for the target use case." -> I understand that, but actual speech/audio processing tasks, such as zero-shot TTS, prompt-instructed TTS, and audio style transfers, are really complicated. Therefore, I doubt that the benchmark results presented in this paper are really useful in selecting a suitable neural speech/audio codec model for such complicated tasks.
- [C4-2] In Section 5: "While discrete representations may currently underperform continuous features on certain discriminative tasks, particularly in low-resource or semantically fine-grained scenarios, recent advances show that they can match or even outperform continuous representations in some applications, such as text-to-speech, especially when trained on large-scale data. " -> How much data can be considered “large-scale”?

[Minor comments]

Please correct the following typos and errors:
- In Section 2.2.1 (SVQ): "t has gained popularity due to ..." -> should be "It has gained ..." ?
- In Section 2.4.1: "... such a SSL model ..." -> "... such an SSL model ..."
- In Section 3.3.2 (Evaluation Setup): "To evaluate the vocal quality of synthesized speech, we use a pretrained UTMOS model (Takaaki et al., 2022)."
  - The term "vocal quality" sounds unclear. Please rewrite it as "speech naturalness."
  - Takaaki is the author's first name. Please modify it as "Saeki et al., 2022."
- Please use "$k$-means" throughout the paper.

It would be helpful for readers if the authors provide samples of generated speech, audio, and music in their personal web page or supplementary material.

**Strengths And Weaknesses:**

[Strengths]
- The proposed taxonomy for audio tokenization techniques is more intuitive and understandable than the existing "semantic vs. acoustic" classification.
- Thorough experimental investigation through a series of various benchmarks and ablation studies provides insights for further development of audio language modeling techniques.

[Weaknesses]
- This paper requires the reader to have advanced knowledge of recent techniques proposed in audio/speech processing, such as self-supervised learning (SSL) for speech/audio signals.
- Experimental setups in benchmarks and ablation studies are not fully justified.
- Although the quantization techniques are important factors that affect the performances of the neural audio codec models, their mathematical formulations are not always provided.

---

> ### Author Response · Authors · 2025-08-01
> **We thank the reviewer for their time and valuable feedback.**
>
> > **The codebook collapse is well-known issue in VQVAE-based generative models ...**
> Codebook collapse is a known issue in neural audio codecs, similar to other VQ-VAE–based models. It occurs when the model selects only a small subset of code vectors during training, leaving the rest inactive. This under-utilization reduces the effective codebook size, implicitly lowering the target bitrate and limiting the model’s ability to represent diverse audio content. As a result, it can negatively affect reconstruction quality.  In response to this comment, we have added a new discussion of this phenomenon and relevant mitigation strategies in Section 2.4.2: Main Training Objectives, under the Vector Quantization (VQ) loss subsection.
> We have added explanations and mathematical formulations to both Section 2.4.2 (Masked Prediction) and Section 2.4.3 (Disentanglement)
>
> > **Please add explanation with mathematical formulations in some parts,..**
> >
> > &nbsp;
> >
> >We have clarified Section 2.4.3 (Disentanglement) by adding equations and additional explanations to better describe the concepts discussed in this section.
> >
> > &nbsp;
> >
> > We thank the reviewer for pointing out the potential confusion in the Adversarial (GAN) paragraph. In the context of adversarial loss, perceptual loss refers to the discriminator losses that focus on feature- and frequency-level rebuilding, firstly proposed in VQGAN, and followed by common audio vocoders. However, we understand the term “perceptual loss” can also refer to the traditional ones, e.g., psychoacoustic-based losses, in a general context. We adjusted our statement in the manuscripts (2.4.2 Main Training Objectives - Adversarial (GAN) ),  and clearly stated that the multi-scale discriminator in GAN helps to capture different structures in the audio signal and enhances the perceptual quality. In the meantime, we avoid directly using the term “perceptual loss” without a definition.
>
> > **The following terms should be explained with a proper definition...**
> We have added definitions for the terms self-supervised learning (SSL) and acoustic language models in both Section 1 (Introduction) and Section 3.3 (Acoustic Language Models Evaluation) to improve clarity for readers unfamiliar with these concepts.
>
> > **Why did the authors choose the tokenizers listed in Tables 2 and 14...**
> We selected the tokenizers presented in Tables 2 and 14 based on the following criteria:
> >
> > &nbsp;
> >
> > - Public availability and reproducibility: We prioritized open-source models with publicly available checkpoints and code to ensure reproducibility.
> >
> > - Quantization diversity: Our goal was to include a representative set of quantization strategies. The selected models cover a range of techniques, including RVQ (EnCodec, DAC), SVQ (WavTokenizer), FSQ (SQ-Codec), KMeans (Discrete WavLM), and semantically distilled methods (SpeechTokenizer, Mimi).
> >
> > - Domain coverage: We aimed to include tokenizers that support multiple domains, such as speech, music, and general audio to enable a more comprehensive evaluation. However, for certain strategies like semantic distillation and K-Means, existing models are limited to specific domains. We discuss this limitation in the Conclusion and Future Directions section.
> >
> > &nbsp;
> >
> > We add clarification on our selection criteria in “Section 3: Benchmark Evaluation”.
>
> > **Why did the authors employ multiple models to estimate the perceptual quality of generated samples? As the UTMOS was developed in 2022, it mi..**
> As noted in [1,2], UTMOS v1 remains a more prevailing method for evaluating reconstruction and speech generation quality. However, we appreciate the recommendation of UTMOS v2, and agree that comparing the results between UTMOS v1 and v2 is valuable. Therefore, we have included UTMOS v2 results in the appendix for completeness and comparison (Additional Evaluation for Reconstructed Audio Quality).
> >
> > &nbsp;
> >
> > [1] Song, Yakun, et al. "MagiCodec: Simple Masked Gaussian-Injected Codec for High-Fidelity Reconstruction and Generation." arXiv preprint arXiv:2506.00385 (2025).
> >
> > [2] Li, Jiaqi, et al. "DualCodec: A Low-Frame-Rate, Semantically-Enhanced Neural Audio Codec for Speech Generation." arXiv preprint arXiv:2505.13000 (2025).

---

> > ### Author Response · Authors · 2025-08-01
> >
> > > **For all experiments please describe the computational resources,..**
> > We have added the computational settings for each experiment in the appendix section ( B Computational Setup), including the number of GPUs used and their device types.
> >
> > > **In Section 3.4: "For real-world applications, we encourage referring to the full benchmark tables and task-specific analyses to identify the most appropriate..**
> > We agree that selecting the best tokenizer for complex real-world tasks can be challenging. However, our benchmark is designed to evaluate tokenizers from multiple perspectives, with each study focusing on a specific aspect (e.g., reconstruction quality, semantic consistency, or downstream task performance).
> > >
> > > &nbsp;
> > >
> > > While our benchmark may not directly cover every advanced use case, it provides a structured foundation to better inform the user regarding the details of each tokenizer. For example, someone interested in emotional TTS could consider both the TTS results and emotion recognition performance from the downstream evaluation to make a more informed decision. Similarly, if the goal is to train a speech-aware language model (i.e., an LLM that takes audio as input and responds in text), one might prioritize tokenizers that perform well in speech LM and downstream semantic tasks. As our results suggest, models trained with SSL features, such as HuBERT, or semantically distilled tokenizers, better capture high-level semantic content even if they do not provide the best reconstruction quality. In some cases, combining multiple tokenizers or representations may be necessary to meet the demands of more complex pipelines. We provide an interactive database of tokenizers with their characteristics on our [project website](https://poonehmousavi.github.io/dates-website/taxonomy)  to make it easier for users to find the tokenizer best suited to their needs.
> >
> > > **In Section 5: "While discrete representations may currently underperform continuous features on certain discriminative tasks..**
> > The definition of “large-scale” depends on both the complexity of the task and the characteristics of the tokenizer. From our observations, tasks like ASR and TTS typically require more than 1,000 hours of training data to achieve reasonable performance, especially when using acoustic tokenizers such as EnCodec or DAC, which are primarily optimized for reconstruction quality. These models tend to be particularly sensitive in low-resource settings. In contrast, semantic tokenizers, such as those trained on/with SSL features (e.g., Discrete WavLM, SpeechTokenizer), show more robustness under limited data conditions. We observed the effect of data scaling in our ASR experiments, where Discrete WavLM achieved a WER of 4.78% on LibriSpeech (960h), 22.0% on Basque (116h), and 58.9% on Welsh (8h) using a BiLSTM head at low bitrate, demonstrating a clear correlation between training data size and performance. This trend holds across other tasks as well. In music genre classification, emotion recognition, and sound event classification, discrete tokenizers degrade more rapidly as the training data size decreases, whereas continuous representations remain more stable. Overall, performance improves consistently with larger datasets, particularly for acoustic tokenizers, making data scale a key factor when selecting or training models for different tasks.
> >
> > > **It would be helpful for readers if the authors provide samples of generated speech..**
> > We have added example samples from our reconstruction experiments, acoustic language models, and ablation study to our  [project website](https://poonehmousavi.github.io/dates-website/samples).
> >
> > > **typos and errors**
> > Thank you for the comments. We have made the requested changes.

---

### Review · Reviewer_ktZN · 2025-07-10

**Summary Of Contributions:**

The paper presents a comprehensive analysis and benchmark for discrete audio tokenizers, which convert audio signals into discrete, compact representations useful for efficient audio processing and multimodal integration. The contributions of the paper include a taxonomy that classifies audio tokenizers by encoder-decoder architecture, quantization techniques, training methods, streamability, and application domains; an extensive benchmark evaluation across multiple audio domains assessing reconstruction quality, downstream task effectiveness, and acoustic language modeling performance; and ablation studies systematically examining how different factors such as quantization methods, sampling rates, and domain-specific training affect performance.

**Audience:**

Yes

**Broader Impact Concerns:**

As a reviewer, I have no significant ethical concerns regarding the broader impact of this work. The paper focuses on the taxonomy, evaluation, and analysis of discrete audio tokenizers, which are foundational tools in audio processing. The authors do not propose new generative models or applications that could directly lead to misuse, such as deepfakes or surveillance.

**Claims And Evidence:**

Yes

**Requested Changes:**

- Instead of fully replacing the semantic vs. acoustic distinction, consider incorporating it into a broader, hierarchical taxonomy to ensure comprehensive coverage and practical utility.
- Include subjective perceptual evaluation (e.g., MOS for text-to-speech tasks) or clearly justify its omission.
- Clarify explicitly the sources and nature of checkpoints and evaluation tools used in benchmarking.
- Explore and suggest additional evaluation metrics beyond FAD and CLAP for a more robust quality assessment.

**Strengths And Weaknesses:**

**Strengths**

- Proposes a systematic and detailed taxonomy, clearly categorizing audio tokenizers by key dimensions, improving conceptual clarity.
- Provides extensive benchmarking across speech, music, and general audio, including reconstruction quality, downstream tasks, and acoustic language modeling.
- Conducts rigorous ablation studies, effectively highlighting trade-offs regarding quantization methods, sampling rates, domain-specific training, and semantic distillation.
- Presents structured visualizations (tables, radar charts), enabling effective comparison of performance.
- Identifies practical trade-offs between reconstruction accuracy and downstream task performance, providing insights valuable for tokenizer design.
- Covers comprehensive downstream applications (discriminative and generative tasks) showcasing broad applicability.
- Evaluates acoustic language modeling tasks, extending analysis beyond traditional benchmarks.
- Highlights practical challenges related to cross-domain generalization, offering useful insights for universal audio model development.
- Provides meaningful future research directions based on comprehensive analyses of current tokenizer limitations and trade-offs.

**Weaknesses**

- Although addressing limitations of the traditional semantic-acoustic dichotomy, the newly proposed taxonomy does not fully replace it; semantic and acoustic token distinctions remain practically significant.
- Relies predominantly on objective metrics, lacking subjective perceptual assessments (e.g., MOS for text-to-speech tasks), limiting comprehensive evaluation of perceptual audio quality.
- Some benchmarking depends on publicly available pretrained checkpoints rather than uniformly trained models, potentially introducing biases or inconsistencies.
- Relatively insufficient discussion on computational complexity, inference speed, and latency, important aspects for practical applications.

---

> ### Author Response · Authors · 2025-08-01
> **We thank the reviewer for their time and valuable feedback.**
>
> > **Instead of fully replacing the semantic vs. acoustic distinction, consider incorporating it into a..**
> While the terms acoustic and semantic are widely used in the literature and have clear practical relevance, we recognize that this traditional division does not always capture the full complexity of modern tokenization methods. In practice, acoustic tokenizers can capture semantic content, and semantic tokenizers have been successfully applied to generative tasks, often blurring the boundary between these two categories. For example, in our benchmark studies, semantic tokenizers like Discrete HuBERT performed well in speech generation, while acoustic tokenizers such as SpeechTokenizer achieved strong results on semantic tasks like ASR. As tokenization approaches continue to evolve, we believe a more fine-grained taxonomy, one that considers architectural choices, quantization methods, training paradigms, streamability, and application domains, better reflect current trends and practical trade-offs. Rather than replacing the acoustic/semantic distinction, our framework is intended to complement and enrich it, providing a more flexible and comprehensive basis for understanding and comparing audio tokenizers.
>
> > **Relies predominantly on objective metrics, lacking subjective perceptual assessments...**
> We acknowledge that subjective perceptual evaluation, such as MOS tests, plays an important role in assessing audio quality.  However, we would like to highlight that our primary goal is to build a scalable and extensible benchmark. Subjective evaluations are difficult to reproduce consistently over time, as they depend on factors such as participant pool, testing conditions, and timing [1,2]. For this reason, objective evaluation remains the main metric for generative audio models. Although objective scores may not perfectly capture human perception, they still offer a useful and reproducible estimate. Following common practice, and to ensure fairness and replicability, especially as new tokenizers are added, we rely on objective metrics that can be automatically computed across all models. This approach keeps the benchmark consistent, reproducible, and suitable for ongoing, leaderboard-based community evaluation.
> >
> > &nbsp;
> >
> >[1] Cooper, Erica, et al. "Generalization ability of MOS prediction networks." ICASSP 2022-2022 IEEE International Conference on Acoustics, Speech and Signal Processing (ICASSP). IEEE, 2022.
> >
> >[2] Huang, Wen-Chin, et al. "The voicemos challenge 2022." arXiv preprint arXiv:2203.11389 (2022).
>
>
> > **Some benchmarking depends on publicly available pretrained checkpoints rather..**
> We would like to clarify the distinction between the two parts of our study: Section 3 (Benchmark Evaluation) and Section 4 (Ablation Studies).
> >
> > &nbsp;
> >
> > Section 3 is based on publicly available pretrained checkpoints and aims to evaluate how existing tokenizers perform across a range of tasks (e.g., AudioLM, ASR, transmission) from multiple perspectives. The focus is on identifying the best use cases for these models and understanding how to use them effectively in real-world applications. In many practical scenarios, users do not train new tokenizers from scratch but instead select a suitable pretrained tokenizer, either as a feature extractor or a vocoder, based on their target use case. Our benchmark study is designed to address this common need by providing guidance on the relative strengths of existing models.
> >
> > &nbsp;
> >
> >Section 4, on the other hand, presents controlled ablation studies using uniformly trained tokenizers. This section explores best practices for training new tokenizers, including architectural choices and training strategies. It is aimed at researchers or practitioners interested in building and optimizing their own tokenizer models.
> >
> > &nbsp;
> >
> >By separating these two studies, we provide both practical insights for using existing tools and foundational guidance for developing new ones.

---

> > ### Author Response · Authors · 2025-08-01
> > **Addressing remaining points**
> >
> > > **Relatively insufficient discussion on computational complexity, inference...**
> > Table 2 in Section 3.1 (Evaluation for Reconstructed Audio Quality and Complexity) compares tokenizers in terms of number of parameters, MACs, frame rate, and token rate(Table 4), all of which are key factors influencing inference speed and latency. To address this comment, we have added the computational settings for each experiment in Appendix B (Computational Setup), including details on the approximate training time, number, and type of GPUs used.
> >
> > > **Clarify explicitly the sources and nature of checkpoints...**
> > We have listed the sources of the tokenizer checkpoints used in our benchmark studies in Table 2. For each benchmark section, we also specify the tools and metrics used for evaluation. Additional details are illustrated in Figure 1 and discussed in the Contributions section.
> >
> > > **Explore and suggest additional evaluation metrics beyond FAD...**
> > We follow the common practice for evaluation of text-to-audio generation by reporting three complementary evaluation metrics - FAD, KL-Divergence and CLAP score [1-5]. We do note that better metrics would be of great use to the audio research community in sub-section 3.3 (Evaluation Setup and Summary). However, if the reviewer has any specific suggestions for metrics that would improve the clarity of the results and trends, we are happy to consider adding them.
> > >
> > > &nbsp;
> > >
> > > [1] Kreuk, Felix, et al. "AudioGen: Textually Guided Audio Generation." The Eleventh International Conference on Learning Representations.
> > >
> > > [2] Ziv, Alon, et al. "Masked Audio Generation using a Single Non-Autoregressive Transformer." The Twelfth International Conference on Learning Representations.
> > >
> > > [3] Evans, Zach, et al. "Stable audio open." ICASSP 2025-2025 IEEE International Conference on Acoustics, Speech and Signal Processing (ICASSP). IEEE, 2025.
> > >
> > > [4] Yao, Yao, et al. "Jen-1 composer: A unified framework for high-fidelity multi-track music generation." Proceedings of the AAAI Conference on Artificial Intelligence. Vol. 39. No. 13. 2025.
> > >
> > > [5] Jung, Jaemin, et al. "Voicedit: Dual-condition diffusion transformer for environment-aware speech synthesis." ICASSP 2025-2025 IEEE International Conference on Acoustics, Speech and Signal Processing (ICASSP). IEEE, 2025.

---

### Review · Reviewer_YDdA · 2025-07-21

**Summary Of Contributions:**

This paper presents a comprehensive review and benchmark of discrete audio tokenizers.

- It introduces a fine-grained taxonomy to categorize tokenizers across multiple dimensions, including encoder-decoder architecture, quantization method, training paradigm, streamability, and application domains.

- It conducts thorough benchmarking from three key perspectives: reconstruction quality, downstream task performance, and acoustic modeling/generation capabilities.

- This study further investigates the impact of factors such as training data domain, sampling rate, semantic distillation, and quantization techniques on tokenizer performance through ablation studies by isolating relevant components.

**Audience:**

Yes

**Broader Impact Concerns:**

No broader impact concerns.

**Claims And Evidence:**

Yes

**Requested Changes:**

Below are some of my detailed comments and questions.
- In Section 2.4.2, for the Vector Quantization section, when presenting the training objective of vector quantization, the authors could add reference for the VQ-VAE work [1].
- In Section 3.1, General Audio and Music section, it writes “SQ-Codec, the only fully out-of-domain model (trained only on speech data), gives the poorest performance in SDR and SI-SNR metrics.” However, as illustrated in Table 5, it seems that the WT-S-24 model is the out-of-domain model and achieves the poorest performance in SDR and SI-SNR metrics.
- In Section 3.2, Results and Discussion for Audio and Music Tasks, the paper states: 'Even the best-performing model (EnCodec at medium bitrate) only reaches about -7 dB SI-SDR for audio and -5.7 dB for music.' However, Table 9 reports that EnCodec at medium bitrate (#Q=8) achieves 9.53 dB SI-SDR for audio and 1.98 dB for music. Could the authors clarify whether I have misunderstood this, or if there is a discrepancy between the statement and the table?
- In Section 3.2, Summary, the authors discuss the performance of semantic and acoustic tokenizers. However, the paper does not clearly indicate which benchmarked methods fall under each category. It would be helpful if the authors could clarify this distinction.
- In Section 5, the authors discuss semantic distillation beyond speech. However, it is unclear to me how 'semantic' is defined in the context of music and general audio. While this may be an open question, I would appreciate hearing the authors' perspective on this.

[1]Van Den Oord, Aaron, and Oriol Vinyals. "Neural discrete representation learning." In Advances in neural information processing systems, 2017.

**Strengths And Weaknesses:**

Strengths:
- This work presents a clear and well-defined taxonomy for classifying state-of-the-art discrete audio tokenizers, offering the community a structured framework for comparing and analyzing their different components.
- This work thoroughly benchmarks a range of tokenizers through extensive experiments. The benchmarking tasks are organized into three categories, each reflecting a distinct aspect of model capacity and performance. Task descriptions, evaluation datasets, setups, and metrics are clearly detailed, and the results yield valuable insights into the strengths and limitations of different approaches.
- The investigation covers a wide range of application domains—including speech, music, and general audio—and also illustrates the domain adaptation and generalization capabilities of different tokenizers.
- The paper is clearly written and well-organized, with figures and tables effectively illustrating the content.

Overall, this is a high-quality piece of work that presents a thorough investigation of current discrete audio tokenizers, offering a comprehensive empirical understanding of state-of-the-art approaches. I am confident that this work will contribute significantly to advancing research and development in the field.

Weakness:
- This work lack a discussion of the training datasets and data volume, which are likely key factors influencing the model’s performance.

---

> ### Author Response · Authors · 2025-08-01
> **We thank the reviewer for their time and valuable feedback.**
>
> > **Lacks a discussion of the training datasets**
> We describe the datasets and their specifications for each experiment in the "Dataset" subsection of the relevant sections. In response to your comment, we have also added a summary table in Appendix C (“Dataset”) that lists the datasets used for each experiment.
>
> > **In Section 2.4.2, for the Vector Quantization..**
> We have  referred to this paper in this section:
> “In the straight-through estimator (van den Oord et al., 2017) used for vector
> Quantization, gradients bypass the codebook.”
>
> > **In Section 3.1, General Audio and Music section, it writes “SQ-Codec,...**
> Thank you for pointing this out. This was indeed a mistake on our part. We intended to refer to WT-S-24 as the out-of-domain model with the lowest SDR and SI-SNR performance. We have corrected this error in the revised version of the paper (see Section 3.1, Page 17).
>
> > **In Section 3.2, Results and Discussion for Audio and Music Tasks, the paper states: 'Even the best-performing model...**
> We report the performance in terms of SI-SDR improvement (“SI-SDRi”), not absolute SI-SDR. Thus, the reason we also provide the performance on “unprocessed” mixtures. For instance, for general audio, we report an improvement of 9.53 dB over the mixture (-16.5 dB). In absolute value, this means that the resulting predictions yield an average of -7-7 dB performance. We clarified this point in Section 3.2, Results and Discussion for Audio and Music Task.
>
> > **In Section 3.2, Summary, the authors discuss the performance of semantic and acoustic tokenizers...**
> We have clarified this distinction by explicitly listing the tokenizers under each category in Section 3.2 (Summary). Specifically, we refer to Discrete WavLM as a semantic tokenizer (using an SSL model in a post-training setup), Mimi and SpeechTokenizer as semantically distilled tokenizers, and EnCodec, DAC, WavTokenizer, and SQ-Codec as acoustic tokenizers.
>
> > **In Section 5, the authors discuss semantic distillation beyond speech...**
> When applying semantic distillation to speech, we typically refer to methods that distill information from a pretrained self-supervised learning model into the tokenizer’s training process. For example, models like Mimi and SpeechTokenizer use SSL features (e.g., from WavLM or HuBERT) to guide specific RVQ layers, encouraging them to learn representations aligned with the phonetic content encoded by the SSL model. However, in the domains of music and general audio, there are currently no tokenizers that apply semantic distillation in the same way, i.e., using pretrained SSL models such as MERT [1] (for music) or BEATs [2] (for general audio) to guide the tokenizer. In this context, semantic refers to leveraging SSL models to capture coarse-grained, high-level features [3][4] learned from unsupervised large-scale audio data. We acknowledge this as an open question and have highlighted it in Section 5 as a promising direction for future research.
> >
> > &nbsp;
> >
> >[1] Li, Yizhi, et al. "Mert: Acoustic music understanding model with large-scale self-supervised training." arXiv preprint arXiv:2306.00107 (2023).
> >
> >[2] Chen, Sanyuan, et al. "Beats: Audio pre-training with acoustic tokenizers." arXiv preprint arXiv:2212.09058 (2022).
> >
> > [3] Sager, Sebastian, et al. "Audiosentibank: Large-scale semantic ontology of acoustic concepts for audio content analysis." arXiv preprint arXiv:1607.03766 (2016).
> >
> >[4] Sager, Sebastian, et al. "Audiosentibank: Large-scale semantic ontology of acoustic concepts for audio content analysis." arXiv preprint arXiv:1607.03766 (2016).

---

### Author Response · Authors · 2025-08-01
**General Points**

We thank the reviewers for their feedback.  We are glad to see that all reviewers unanimously recognized the diversity and comprehensiveness of our survey and benchmark. We have uploaded a revised version of the paper in PDF format, incorporating the changes requested by the reviewers. All modifications are highlighted in blue for clarity. In addition to the detailed replies to each reviewer, we would like to address some common points raised and provide updates on the actions we’ve taken.

> **Adding computational resource**
We have added the computational settings for each experiment in the appendix section ( Appendix B:  Computational Setup), including the number of GPUs used and their device types.

> **Adding extra metric for Speech reconstruction**
We have included UTMOS v2 results in the appendix for completeness and comparison (Appendix A: Additional Evaluation for Reconstructed Audio Quality).

> **Listenable Samples**
We have added example samples from our reconstruction experiments, acoustic language models, and ablation study to  our [project website](https://poonehmousavi.github.io/dates-website/samples)

---

### Decision · Action_Editor_RXuv · 2025-08-29

**Recommendation:** Accept as is

**Additional Comments:**

This paper provides a comprehensive review and benchmarking of discrete audio tokenizers.
Several concerns were raised by reviewers, including the following: insufficient references to research on vector quantization; a lack of theoretical formalization, such as masking prediction and perceptual loss in GANs; the need for a hierarchical taxonomy that goes beyond the semantic/acoustic dichotomy; transparency in experimental design and evaluation; and unclear computational resource requirements.
The authors addressed these concerns and made the necessary revisions.
These responses and revisions resolved the concerns. Finally, the reviewers decided to accept the paper, and the AE agrees.

**Audience:**

Yes

**Audience Explanation:**

Audio information processing is one of the key application areas for machine learning. Regarding discrete audio tokenizers, this paper organizes the advantages, limitations, and trade-offs of each method based on experimental evidence. These insights provide valuable guidelines for future research. This research is expected to attract the interest of many researchers involved in audio information processing using machine learning.

**Claims And Evidence:**

Yes

**Claims Explanation:**

This paper provides a thorough review and benchmarking of discrete audio tokenizers. Specifically, it organizes a taxonomy based on multidimensional perspectives, including encoder-decoder architecture, quantization techniques, learning paradigms, streamability, and application domains.

Furthermore, it presents extensive experiments based on multiple benchmarks and key evaluation criteria, such as reconstruction quality, downstream task performance, and acoustic modeling capability. Ablation studies examine the effects of various factors, such as quantization schemes, sampling rates, and domain specificity, lending precision and persuasiveness to the paper's claims.